# GRAPH BACKUP: DATA EFFICIENT BACKUP EXPLOITING MARKOVIAN TRANSITIONS

## ABSTRACT

The successes of deep Reinforcement Learning (RL) are limited to settings where we have a large stream of online experiences, but applying RL in the data-efficient setting with limited access to online interactions is still challenging. A key to data-efficient RL is good value estimation, but current methods in this space fail to fully utilise the structure of the trajectory data gathered from the environment. In this paper, we treat the transition data of the MDP as a graph, and define a novel backup operator, Graph Backup, which exploits this graph structure for better value estimation. Compared to multi-step backup methods such as $n$-step $Q$-Learning and TD($\lambda$), Graph Backup can perform counterfactual credit assignment and gives stable value estimates for a state regardless of which trajectory the state is sampled from. Our method, when combined with popular off-policy value-based methods, provides improved performance over one-step and multi-step methods on a suite of data-efficient RL benchmarks including MiniGrid, Minatar and Atari100K. We further analyse the reasons for this performance boost through a novel visualisation of the transition graphs of Atari games.

## 1 INTRODUCTION

Deep Reinforcement Learning (DRL) methods have achieved super-human performance in a varied range of games (Mnih et al., 2015; Silver et al., 2016; Berner et al., 2019; Vinyals et al., 2019). All of these present a proof of existence for DRL: with a large amount of online interaction, DRL-trained policies can learn to solve problems that have similar properties to real-world decision-making tasks. However, most real-world tasks such as autonomous driving or financial trading are hard to simulate, and generating new interaction data can be expensive. This makes it crucial to develop data-efficient RL approaches that solve sequential decision-making problems with limited online environment interactions.

As many existing DRL algorithms assume access to a simulator they don't focus on efficiently using the available data as it's always cheaper to simply generate fresh data from the simulator. Data is normally stored in a buffer and only used several times for learning before being discarded. However, there is lots of additional structure in the transition data, and a key insight of our work is to organise the trajectories stored in the buffer as a graph (For example see Figure 1(a) which shows a visualisation of the transition graph of the Atari game Frostbite). Our method, Graph Backup, then exploits this transition graph to provide a novel backup operator for bootstrapped value estimation. When estimating the value of a state, it will combine information from a subgraph rooted at the target state, including rewards and value estimates for future states.

When the environment has Markovian transitions and crossovers between trajectories, the construction of this data graph provides several benefits. As discussed in Section 4.2, our method exploits intersecting trajectories to correctly propagate reward to more states, effectively by propagating reward along an imagined trajectory. Further, while existing improvements to one-step backup (as used in by Mnih et al. (2015)) such as multi-step backup (Moriarty & Miikkulainen, 1995; Hessel et al., 2018; Sutton & Barto, 2018) address the problem of slow reward information propagation (Hernandez-Garcia & Sutton, 2019), they add variance to the state value estimates as different states can have different values estimates depending on the trajectory they were sampled from. Our method addresses this issue by grouping states in the transition graph and averaging over outgoing transitions at the value estimation stage.

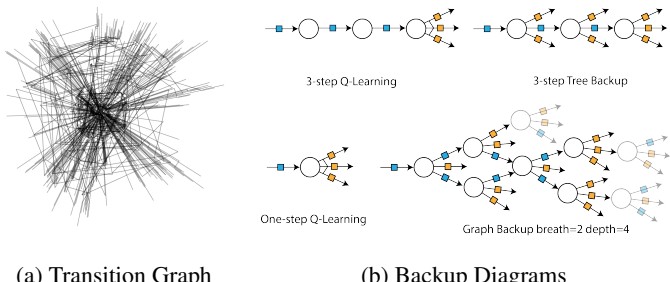

(a) Transition Graph   (b) Backup Diagrams

Figure 1: (a) shows the transition graph Frostbite, an Atari game, extracted from a replace buffer of a Graph Backup agent after 100k steps. (b) shows backup diagrams for different backup targets. The circles are states, the blue squares represent the actions that have been observed for the given state node, and orange squares are actions where target net evaluation happened

We propose a specific implementation of Graph Backup, extending Tree Backup (Precup et al., 2000) (Section 4, see Figure 7(c)). Our method improves data efficiency and final performance on MiniGrid (Chevalier-Boisvert et al., 2018), Minatar (Young & Tian, 2019) and Atari100K when using Graph Backup combined with DQN (Mnih et al., 2015) and Data-Efficient Rainbow (van Hasselt et al., 2019) compared to other backup methods, showing that utilising the graph structure of the trajectory data leads to improved performance in the data-efficient setting (Section 5). To more fully understand where this gain in performance comes from, we further investigate the graph sparsity of different environments in relation to the performance of Graph Backup, in part using a novel method to visualise the full set of seen transitions and their graph structure (Section 6).

## 2 RELATED WORK

The idea of multi-step backup algorithms (e.g. TD($\lambda$), $n$-step TD) dates back to early work in tabular reinforcement learning (Sutton, 1988; Sutton & Barto, 2018). Two approaches to multi-step targets are $n$-step methods and eligibility trace methods. The $n$-step method is a natural extension of using a one-step target that takes the rewards and value estimations of $n$ steps into future into consideration. For example, the $n$-step SARSA (Rummery & Niranjan, 1994; Sutton & Barto, 2018) target for step $t$ is simply the sum of $n$-step rewards and the value at timestep $t + n$: $R_{t+1} + R_{t+2} + ... + R_{t+n-1} + V(S_{t+n})$. Graph Backup is an extension of an $n$-step backup target, Tree Backup, which will be described in Section 3.

*Eligibility trace* (Sutton, 1988) methods instead estimate the $\lambda$-return, which is an infinite weighted sum of $n$-step returns. The advantage of the eligibility trace method is it can be computed in an online manner without explicit storage of all the past experiences, while still computing accurate target value estimates. However, in the context of off-policy RL, eligibility traces are not widely applied because the use of a replay buffer means all past experiences are already stored. In addition, eligibility traces are designed for the case with a linear function approximator, and it's nontrivial to apply them to neural networks. van Hasselt et al. (2021) proposed an extension of the eligibility trace method called *expected eligibility traces*. Similar to Graph Backup, this allows information propagation across different episodes and thus enables counterfactual credit assignment. However, similar to the original eligibility traces methods, it is a better fit for the linear and on-policy case, whereas Graph Backup is designed for the non-linear and off-policy cases.

Since a learned model can be treated as a distilled replay buffer (van Hasselt et al., 2019), we can view model-based reinforcement learning as related to our work. Recent examples include Schrittwieser et al. (2020); Hessel et al. (2021); Farquhar et al. (2018); Hafner et al. (2021b); Kaiser et al. (2020b); Ha & Schmidhuber (2018). These MCTS-based algorithms also share some similarities with Graph Backup as they also utilise tree-structured search algorithms. However, our work is aimed at model-free RL, and so is separate from these works.

Several recent works have also utilised the graph structure of MDP transition data. Zhu et al. (2020) propose to use the MDP graph as an associative memory to improve Episodic Reinforcement Learning

(ERL) methods. This allows counterfactual reward propagation and can improve data efficiency. However, the usage of a data graph in this work is different from the usage in Graph Backup: the graph is used for control and as an auxiliary loss, rather than for target value estimation. Their associative memory graph also doesn't handle stochastic transitions and the return for each trajectory is only based on observed return (no bootstraping is used), unlike our work. Topological Experience Replay (Hong et al., 2022, TER) uses the graph structure of the data in RL for better replay buffer sampling. TER uses the graph structure to decide which states should be sampled from the replay buffer during learning, by implementing a sampling mechanism that samples transitions closer to the goal first. This work is orthogonal (and possibly complementary) to ours, as TER is a replacement for uniform or prioritized sampling from a replay buffer while Graph Backup is a replacement for one-step or multistep backup for value estimation.

## 3    PRELIMINARIES: ONE-STEP AND MULTI-STEP BACKUP

Given an MDP $\mathcal{M}$ we denote $\mathcal{A}$ as the action space; $\mathcal{S}$ to be state space; $\mathcal{R} \subset \mathbb{R}$ to be reward space; and $a_t \in \mathcal{A}$, $s_t \in \mathcal{S}$ are used to denote the specific actions and states respectively observed at step $t$. We denote a trajectory of states, actions and rewards as $\tau = (s_1, a_1, r_1, s_2, a_2, r_2, ...)$.

For a transition $(s_t, a_t, r_t, s_{t+1})$ the loss function of DQN methods is defined as the mean square error[1] between the predicted $q$-value and the backup target $G^{a_T}$ for $(s_t, a_t)$:

$$L(\boldsymbol{\theta}|s_t, a_t) \stackrel{\text{def}}{=} \left(q_{\boldsymbol{\theta}}(s_t, a_t) - G^{a_t}\right)^2,  \tag{1}$$

where $q_{\boldsymbol{\theta}}$ represents the online network parameterized by $\boldsymbol{\theta}$. The backup target $G^{a_t}$ is an estimation of the optimal Q-value $q^*(s_t, a_t)$. Vanilla DQN uses one-step bootstrapped backup, which makes gradient descent an analogue to the update of tabular $Q$-learning:

$$G_{t:t+1}^{a_t} \stackrel{\text{def}}{=} r_{t+1} + \gamma \max_{a'} q_{\boldsymbol{\theta}'}(s_{t+1}, a')  \tag{2}$$

where $\theta'$ are the parameters of the target network, which is standard in DQN.

The one-step target makes the propagation of the reward information to previous states slow, which is amplified by the use of a separate frozen target network. This motivates the use of more sample-efficient multi-step targets in DQN (Hessel et al., 2018; Hernandez-Garcia & Sutton, 2019).

A widely used multi-step backup algorithm is $n$-step $Q$-Learning ($n$-step-$Q$) (Hessel et al., 2018; Silver et al., 2017). This method sums the rewards in next $n$ steps, together with the maximum $q$ value at step $n$:

$$G_{t:t+n}^{a_t} \stackrel{\text{def}}{=} r_{t+1} + \gamma r_{t+2} + ... + \gamma^n \max_{a'} q_{\boldsymbol{\theta}'}(s_{t+n}, a').  \tag{3}$$

$n$-step-$Q$ exploits the chain structure of the trajectories with little computational cost but at a cost of biased target estimation. The distribution of the sum of the rewards $r_{t+1} + \gamma r_{t+2} + ... + \gamma^{n-1} r_{t+n}$ are conditioned on the behaviour policy $\mu$ which generates the data. This means that in an off-policy setting the estimated target value can be biased towards the value of the behaviour policy.

Another off-policy multi-step target is Tree Backup (Precup et al., 2000). Tree Backup is designed for general-purpose off-policy evaluation, meaning it aims to estimate the value of any target policy $\pi$ by observing the behaviour policy $\mu$. When the target policy is the optimal policy given by $q_{\boldsymbol{\theta}'}$, Tree Backup recursively applies one-step-$Q$ backup to the trajectory, bootstrapping with the target value network when the input action $a$ isn't that taken in the trajectory ($a_t$):

$$G_{t:t+n}^{a_t} \stackrel{\text{def}}{=} \begin{cases} r_{t+1} + \gamma \max_{a'} G_{t+1:t+n}^{a'}, & \text{if } t < n, a = a_t \\ q_{\boldsymbol{\theta}'}(s_t, a), & \text{otherwise.} \end{cases}  \tag{4}$$

Despite what its name suggests, Tree Backup does not expand a tree of states and transitions, and so still only leverages the chain structure of trajectories. The name is because the trajectory has leaves corresponding to the actions that were not selected in the current trajectory. In Figure 7(c) we show the backup diagram of the 3-step Tree Backup, where yellow squares are these leaf actions.

---

[1]Or sometimes the Huber Loss (Huber, 1992)

## 4  GRAPH BACKUP

In this section we introduce a new graph-structured backup operator, Graph Backup, extending the multi-step method Tree Backup. Graph Backup allows counterfactual reward propagation and variance reduction while also having the benefits of multi-step backup.

### 4.1  INTRODUCING GRAPH BACKUP

We propose the Graph Backup operator that propagates temporal differences across the whole data graph rather than a single trajectory. The differences between one-step, multi-step, tree and Graph backup are illustrated in Figure 7(c). We want a backup method that can work with stochastic transitions, which means a single state-action pair can lead to different states. This means it's not obvious how to perform recursive backups to the next state, as there could be multiple next states. We estimate the transition probability to each next state using visitation counts, and use the estimated transition probabilities to compute the empirical mean over all possible state value estimates weighted by the likelihood of transitioning to that state. This is easy to calculate efficiently and provides strong results as seen in Section 5.

Denoting the set of all seen transitions to be $\mathcal{T} \subseteq \mathcal{S} \times \mathcal{A} \times \mathcal{R} \times \mathcal{S}$, a counter function $f : \mathcal{T} \to \mathbb{N}^+$ maps each transition $T = (s, a, r, s')$ to its frequency $f(T)$. Notably, this counter function also plays 2 roles 1) the adjacency list of the graph; 2) to weight transitions when the same action leads to different future states. The Graph Backup target for a state-action pair $(s, a)$ is then the average of recursive one-step backup of all outgoing transitions. Similar to Tree Backup, if the $(s, a)$ has not been seen, the target is estimated directly by the target network. Define $\mathcal{T}_{s,a} \stackrel{\text{def}}{=} \{(\hat{s}, \hat{a}, \hat{r}, \hat{s}') \in \mathcal{T} | \hat{s} = s, \hat{a} = a\}$, the set of all $(\hat{s}, \hat{a}, \hat{r}, \hat{s})$ tuples starting with $s, a$. Extending Tree Backup, we can then define the *Graph Backup* (GB) value estimate as

$$G_s^a \stackrel{\text{def}}{=} \begin{cases} \frac{1}{c(s,a)} \sum_{T \in \mathcal{T}_{s,a}} f(T) \left( \hat{r} + \gamma \sum_{a'} \pi(a'|\hat{s}') G_{\hat{s}'}^{a'} \right) & \text{if } c(s, a) > 0 \\ q_{\boldsymbol{\theta}'}(s, a) \text{otherwise.} \end{cases} \tag{5}$$

where $c(s, a) = \sum_{T \in \mathcal{T}_{s,a}} f(T)$ is the normaliser, $\pi$ is the target policy and $q_{\boldsymbol{\theta}'}$ is the target network. In the case where target policy always chooses the action with optimal Q value $\pi(a|s) = \mathbb{1}(\operatorname{argmax}_{a'} G_s^{a'} = a)$, the formula can be simplified into:

$$G_s^a \stackrel{\text{def}}{=} \begin{cases} \frac{1}{c(s,a)} \sum_{T \in \mathcal{T}_{s,a}} f(T) \left( \hat{r} + \gamma \max_{a'} G_{\hat{s}'}^{a'} \right) & \text{if } c(s, a) > 0 \\ q_{\boldsymbol{\theta}'}(s, a) \text{ otherwise.} \end{cases} \tag{6}$$

This is often the case since our implementations are based on DQN and we are interested in the optimal Q-value. In this paper *Graph Backup* refers to the simplified version in Equation (6). On a high level, Graph Backup does dynamics programming Q evaluations on a Empirical MDP, where all the Q values for untried state-action pairs are initialized by a Q network.

Our Graph Backup implementation extends Tree Backup. However, there could be other implementations which extend other multi-step methods, such as $n$-step-$Q$ backup or the $n$-step version (Hernandez-Garcia & Sutton, 2019) of Retrace (Munos et al., 2016). In Appendix I, we present a variation of Graph Backup that extends $n$-step-$Q$ backup.

Note that in Equations (6), (7) and (8), the graph structure does not appear explicitly. This is because it's easier to mathematically formalise these backup operators using transition counts; from an implementation perspective building and maintaining the data graph is the most efficient way of calculating these target value estimates. To better provide intuition for Graph Backup, in Appendix C we explicitly describe the data graph generated from an MDP and link that to Equation (6). The data graph contains the information for calculating and sampling from $\mathcal{T}, \mathcal{T}_s, \mathcal{T}_{s,a}, c(s, a), c(s)$ and $f(T)$.

### 4.2  ADVANTAGES OF GRAPH BACKUP

In Figure 2 we explain conceptually how Graph Backup brings benefits to value estimation and thus the learning of the agent. We also present a more empirical analysis in Appendix M Figure 7. Assuming the value estimates of all the states are initialised as 0, the one-step backup can update

the value of only 1 state. The multi-step backup methods can further propagate the reward to the whole trajectory that leads to the reaching of the goal.[2] However, Graph Backup goes beyond that and propagates rewards to the states of another trajectory (the dashed line). This feature of counterfactual reward propagation can significantly benefit the credit assignment of sparse reward tasks: During the exploration of a sparse-reward environment, policies usually generate a large number of trajectories that do not reach the goal, and while multi-step methods cannot efficiently leverage those transitions, Graph Backup can reuse them by propagating rewards from the crossovers with other successful trajectories.

The second row of Figure 2 shows another advantage of Graph Backup: reducing the variance of the value estimate. Multi-step backup in this case will assign different value estimates for the same state depending on the trajectory the state is sampled from (as it will appear multiple times in the replay buffer). This brings extra noise to the value estimate which can be harmful to learning.[3] In Figure 5, we showed a simple case in MiniGrid where this target value noise can constantly disturb the convergence of DQN. Graph Backup removes this source of variance by ensuring that the same state always has the same value estimate regardless of which trajectory it's sampled from by calculating the value estimate from the underlying data graph. In addition, in stochastic environments Graph Backup reduces variance by averaging over different next states from the same state-action pair.

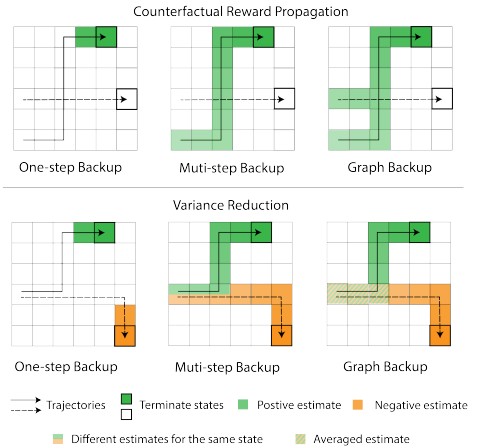

Figure 2: Benefits of Graph Backup

### 4.3 LIMITING EXPANSION OF THE GRAPH

A naïve implementation of Graph Backup would follow the definition exactly and do an exhaustive recursive expansion of the graph. However, the computational cost of doing so can quickly blow up with the size of the replay data.[4] Therefore, similar to the $n$-step backup methods, we need to limit recursive calls. For Graph Backup, this means expanding a smaller local graph from the source state, using the target network for value estimation when reaching expansion limits. In our work, the expansion of the local data graph has both a breadth limit $b$ and a depth limit $d$. When the breadth limit is hit ($|\mathcal{T}_{s,a}| > b$), we will sample $b$ transitions from $\mathcal{T}_{s,a}$ according to their frequency $f$, as opposed to expanding all transitions. If the depth limit is hit ($d < n$) the expansion of the graph will be terminated (so the second case in Equations (6) and (7) is taken).

The pseudocode for local graph expansion is shown in Algorithm 2. Figure 7(c) also illustrates examples of limited expansion for Graph Backup, where faded nodes are clipped away due to hitting the limit.

---

[2]In this case, both Tree Backup and $n$-step-$Q$ backup can produce the estimation shown in the example tasks.

[3]In the Figure 2, the noise comes from different rewards at the end of the trajectories.

[4]In fact, if there are loops in the graph, the situation can be even worse as the algorithm may never converge.

In our work, we make sure the expansion will reach $d$ steps in order to better align with multi-step methods. This makes sure the algorithm will reduce to Tree Backup gracefully when there are no crossovers between the trajectories. It also allows a more principled comparison between Tree Backup and Graph Backup. For $b = 1$ Graph Backup will do a similar job as $d$-step Tree Backup, and increasing $b$ will gradually make the Graph Backup leverage more structure from the transition graph.

## 4.4 INTEGRATION OF OTHER RAINBOW COMPONENTS

To demonstrate that Graph Backup improves data efficiency in a realistic state-of-the-art algorithm, we integrate Graph Backup inside Rainbow (Hessel et al., 2018). As a replacement for $n$-step-$Q$ backup, Graph Backup is orthogonal to all other ingredients. While some components such as prioritized experience replay(PER) (Schaul et al., 2016), noisy networks (Fortunato et al., 2018) and duelling network architectures (Wang et al., 2016) can be plugged in seamlessly, others require more care, which we describe here.

Combining double DQN (van Hasselt et al., 2016) with Tree Backup and Graph Backup is quite straightforward. Double DQN uses an online network instead of a target network to specify the optimal policy in the bellman update, so that $\max_a q_{\boldsymbol{\theta}'}(s, a) = q_{\boldsymbol{\theta}'}(s, \arg\max_a q_{\boldsymbol{\theta}'}(s, a))$ becomes $q_{\boldsymbol{\theta}'}(s, \arg\max_a q_{\boldsymbol{\theta}}(s, a))$ in one-step or $n$-Step-Q backup. For Tree Backup and Graph Backup, we can take the same approach for every expanded state.

Distributional RL (Bellemare et al., 2017), specifically C51, attempts to model the whole distribution of the state-action value rather than the expectation, using a distributional version of the bellman update (namely, one-step backup) when applied in the DQN setting. C51 divides the support of the value into discrete bins, called atoms, and the q network then outputs categorical probabilities over the atoms. In the distributional bellman update, the vanilla bellman update is applied to each atom, and the probability of the atom is distributed to the immediate neighbours of the target value. The loss is the KL divergence between the target and predicted value distribution rather than the mean squared error. In order to combine C51 and Tree Backup or Graph Backup, we apply the distributional bellman update in every state node.

---

**Algorithm 1** Double Distributional Graph Backup

**Input:** source state $S_{\text{source}}$, source action $A_{\text{source}}$, frequency mapping $f : \mathcal{T} \to \mathbb{N}^+$, list of states in the subgraph $L$, atoms $z_0, z_1, ..., z_{N-1}$, online network $p(\cdot, \cdot|\boldsymbol{\theta})$ and target network $p(\cdot, \cdot|\boldsymbol{\theta}')$
1: Set $S_{\text{expanded}}$ be the set containing all the states in list $L$
2: Initialize the target values $\bar{G}_s^a = q_{\boldsymbol{\theta}'}(s, a), \forall s \in S_{\text{expanded}}, a \in \mathcal{A}$
3: **for** $(s, a)$ in $l_{\max}, l_{\max\text{-}1}, ..., l_1$ **do**
4:     $a^* = \arg\max_a \sum_i z_i p_i(s, a|\boldsymbol{\theta})$
5:     $m_i(s, a) = 0, \quad i \in 0, 1, ..., N - 1$
6:     **for** $j \in 0, 1, ..., N - 1$ **do**
7:         **for** $t = (s, a, r, s') \in \mathcal{T}_{s,a}$ **do**
8:             $z'_j \leftarrow [r + \gamma z_j]_{V_{\text{MIN}}}^{V_{\text{MAX}}}$
9:             $b_j \leftarrow (z'_j - V_{\text{MIN}})/\Delta z$
10:             $l \leftarrow \lfloor b_j \rfloor, u \leftarrow \lceil b_j \rceil$
11:             $m_l(s, a) \leftarrow m_l(s, a) + \frac{f(s,a,r,s')}{c(s,a)} p_j(x_{x+1}, a^*)(u - b_j)$
12:             $m_u(s, a) \leftarrow m_u(s, a) + \frac{f(s,a,r,s')}{c(s,a)} p_j(x_{x+1}, a^*)(b_j - l)$
13:         **end for**
14:     **end for**
15: **end for**
16: **return** $m_0(S_{\text{source}}, A_{\text{source}}), ..., m_{N-1}(S_{\text{source}}, A_{\text{source}})$

---

In Algorithm 1, we combine double and distributional RL with Graph Backup given the subgraph state list calculated by Algorithm 2. Blue lines show the changes introduced by Graph Backup.

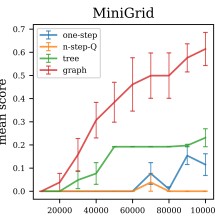 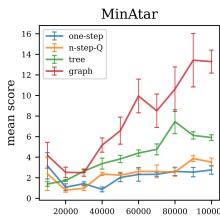 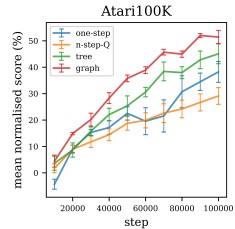 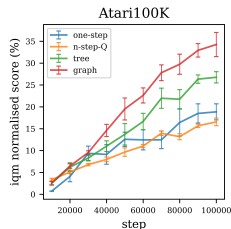

Figure 3: Summary of training curve for Minigrid, Minatar and Atari100K. For Atari100K, we show both the mean and IQM of the human-normalised scores.

Table 1: Numeric summary of the performance. IQM stands for interquartile mean (Agarwal et al., 2021).

|  | one-step | $n$-step-$Q$ | Tree | Graph |
|---|---|---|---|---|
| MiniGrid-IQM | 0.0 | 0.0 | 0.0 | **0.74** |
| MiniGrid-mean | 0.14 | 0.02 | 0.2 | **0.58** |
| MiniGrid-median | 0.0 | 0.0 | 0.0 | **0.58** |
| MinAtar100K-IQM | 1.90 | 1.72 | 3.20 | **8.46** |
| MinAtar100K-mean | 3.07 | 3.76 | 6.26 | **11.83** |
| MinAtar100K-median | 2.07 | 1.56 | 3.33 | **4.95** |
| Atari100K-IQM | 18.85 | 16.55 | 26.76 | **34.25** |
| Atari100K-mean | 32.8 | 28.72 | 43.62 | **50.49** |
| Atari100K-median | 13.39 | 16.89 | 23.74 | **30.07** |

## 4.5 ASSUMPTIONS

The effectiveness of Graph Backup relies on two assumptions about the environment: (1) the transition function of the environment is Markovian, and (2) there are crossovers between state trajectories. We show in Section 5 that—perhaps counter-intuitively—these assumptions hold frequently enough in high dimensional environments (Atari100 from pixel input) for Graph Backup to differentiate itself from Tree Backup in a statistically significant manner. As such, these restrictions are not as strict as may appear, and we further discuss how they can be relaxed in Section 7.

## 5 EXPERIMENTS

In order to test whether Graph Backup can bring benefits to the data efficiency of a DRL agent, we conduct experiments on singleton-MiniGrid, MinAtar and Atari100K. These tasks have an increasingly sparse transition graph so that we can see how many crossovers are needed for Graph Backup to bring significant performance improvements. The baseline agent for MiniGrid and MinAtar is DQN (Mnih et al., 2015) and for Atari100K is Data-Efficient Rainbow (van Hasselt et al., 2019). The average training curves of the different backup methods are shown in Figure 3, where we run each algorithm for 5 random seeds. The performance metric for Atari in the plot is the mean and median of human-normalised scores (%). The final performance for each task and method can be found in Table 1, where we also include both mean and median metrics. The full results of each individual task are shown in Table 3 in Appendix.

**MiniGrid** We first compare the methods in 5 singleton MiniGrid tasks: Empty8x8, DoorKey6x6, KeyCorridorS3R1, SimpleCrossingS9N2 and LavaCrossingS9N2. Every single run (out of 5) has a different but fixed random seed within the whole training process. We set the environment to be fully observable so that the transitions are Markovian. The overall number of possible states is small and the data graph is thus quite dense. The reward of MiniGrid is only given at the end of the episode, which makes credit assignment a critical problem. Among the 5 tasks, one-step backup and Tree Backup only managed to converge within $1e5$ steps for the easiest empty room task. For other tasks with more complex navigation (SimpleCrossing and LavaCrossing) and interaction with objects (DoorKey and KeyCorridor), only the Graph Backup converged this low data regime.

Table 2: Minatar results after 1M steps of training. Values for each individual tasks are mean ± standard error.

|  | one-step | $n$-step-$Q$ | Tree | Graph |
|---|---|---|---|---|
| Minatar-seaquest | 5.98±0.97 | **6.93**±0.72 | 6.57±0.43 | 5.87±0.86 |
| Minatar-breakout | 10.60±1.02 | 9.28±1.48 | 7.05±1.39 | **14.38**±1.38 |
| Minatar-asterix | **14.90**±3.86 | 11.25±1.84 | 7.68±1.81 | 8.02±1.93 |
| Minatar-freeway | 39.93±11.77 | 6.00±2.88 | 18.85±12.10 | **40.95**±12.16 |
| Minatar-space_invaders | 37.53±5.26 | 26.07±1.94 | 33.85±2.29 | **44.40**±4.83 |
| IQM | **15.50** | 9.83 | 8.72 | **16.12** |
| mean | 21.79 | 11.91 | 14.80 | **22.72** |
| median | **14.90** | 9.28 | 7.68 | **14.38** |

**Minatar 100K steps.**  We perform experiments on Minatar. Minatar is a collection of miniature Atari games with a symbolic representation of the objects. The game state is fully specified by the observation of a 10 by 10 image, where each pixel corresponds to an object. We set the overall number of interactions to be 100,000, inspired by the Atari100K benchmark (Kaiser et al., 2020b). We can see in Figure 3 that Graph Backup outperforms Tree Backup, $n$-step-$Q$ backup and one-step backup in terms of mean scores and interquartile mean (IQM), in the data-efficient setting.

**Atari100K**  In order to test if Graph Backup can be applied on tasks with pixel observations, we test it in Atari100K. As suggested by its name, Atari100K limits the number of interactions of Atari games to 100,000, which is equivalent to 2 hours of game-play in Atari. Since the human performance scores reported by Mnih et al. (2015) are also from human experts after 2 hours of game-play, Atari100K is considered as a test-bed for human-level data-efficient learning. We follow the standard frame processing protocol used by Rainbow Hessel et al. (2018) without any other downsampling. The frame is processed into an 84 by 84 greyscale image and the observation is a stack of 4 previous frames, which leads to very sparse transition graphs.[5] The baseline we chose for Atari100K is data-efficient Rainbow van Hasselt et al. (2019), which is a variation of Rainbow that is optimized particularly for Atari100K. Consistent with the results in MiniGrid and MinAtar, Graph Backup performs better than one-step and multi-step methods. The results here show: 1) Graph Backup is robust in terms of bringing orthogonal improvements over other DQN improvements. 2) Graph Backup works for high-dimensional, pixel-based tasks that have sparse transition graphs.

**Minatar 1M steps**  In order to see the asymptotic performance of Graph Backup, in Table 2, we also show the results for 1 million steps of training with hyper-parameters still kept in the data-efficient setting. Interestingly, while Graph Backup still preserves a strong advantage over $n$-step-$Q$ and Tree Backup in terms of aggregated metrics, one-step backup shows comparable performance against Graph Backup with more training. Looking at individual tasks, we find out Tree Backup shows inferior performance compared to one-step backup on Breakout, Asterix and Freeway. From a practical perspective, Minatar 1M results show Graph Backup achieves consistent performance in multiple scenarios while the baselines (one-step backup and Tree Backup) only show their advantage in settings with specific data availability.

## 6  ANALYSIS

**Atari Transition Graphs.**  Here we study why Graph Backup can bring benefits to Atari games. Atari games have pixel-based observations, which have $255^{(210*180*3)}$ combinations if each pixel can take any value independently from other pixels. The first question is then: are there crossovers in just 100K transitions? To analyse the graph density quantitatively, we propose to measure the novel state ratio. The novel state ratio is the ratio between the amount of non-duplicated (i.e. unique) states and the amount of all states that the agent has seen. The novel state ratio will be 1 if there are no overlapping states in the transition graph, in which case Graph Backup reduces to Tree Backup. The average novel state ratio of Atari is 0.927, which means the graphs are usually quite sparse (The average novel state ratios of MiniGrid/MinAtar are 0.006/0.298 respectively). However, if we assume

---

[5]Since the agent interacts with the environment every 4 frames, such preprocessing still assumes the transition to be Markovian.

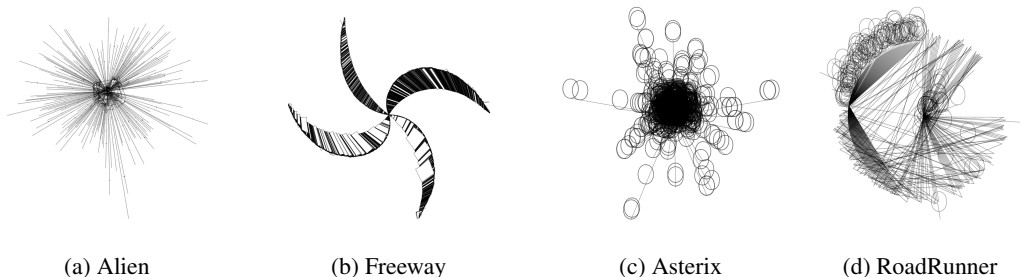

| (a) Alien | (b) Freeway | (c) Asterix | (d) RoadRunner |

Figure 4: Transition graphs of selected Atari100K games, with data collected by a one-step DQN agent. As there are too many state nodes, we did not paint nodes directly but rather preserve edges (transitions), where circles represent self-loop transitions.

that duplicated states happen independently, this means that in 53% of the backup updates, there will be a crossover on the next 10 steps, which means the Graph Backup will give a different value estimate to multi-step methods more than half the time.

However, the crossovers are not distributed uniformly on the transitions graph. In order to investigate the topology of the graph, we visualize the exact graph structure of the whole transition graph. In Figure 4, we show four representative transition graphs and leave the others to Figure 8 in the Appendix. We apply *radial layouts* as proposed by Wills (1999), which scales well with the number of nodes and aligns well with the transition structure of most of the games, where the central point corresponds to the start of the game. Many transition graphs of Atari100k games show interesting crossover structures that can be leveraged by Graph Backup. For example, the transition graph of *Freeway* forms a windmill-like pattern, where each blade corresponds to a group of trajectories that have connections within the group but between groups. There are also some tasks (e.g. Alien) where crossovers mostly happen in the starting stage of the game.

**Explaining Crossovers.** By comparing the transition graphs and the pixel observations, we can provide two explanations for the existence of the trajectory crossovers in Atari games. The first factor is that there is a low number of degrees of freedom for the objects and especially for the agent avatar in many of these games. For most of the Atari games, the agent can only move on a 2D plane, which partially alleviates the curse of dimensionality since we only need two dimensions to specify the position of the agent. A second important factor is that Atari games always have a fixed initial state. Although we follow prior work (Hessel et al., 2018) to add a random number of no-ops after the start of the game, the initial observations the agent sees will still be quite similar, and this creates crossovers at the beginning of the episode (the centre of the plot, for example for Alien in Figure 4).

In light of these characteristics, we recommend practitioners apply Graph Backup with exact state matching to tasks that either (1) have few degrees of freedom, (2) have a discrete state space, or (3) are highly repetitive with minimal noise. There are many real-world tasks that have similar properties, such as any 2D navigation (e.g. household cleaning robots), power management and manufacturing in assembly lines.

## 7    LIMITATIONS AND FUTURE EXTENSIONS

The high-level insight of Graph Backup is to treat all the transition data as a collective entity rather than independent trajectories, and exploit its (graphical) structure for sake of efficient learning. This work shows a simple implementation of this idea, by building the graph with exact state matching. While there are already a large class of tasks that have crossovers, we expect in future to extend Graph Backup to cover even more challenging tasks. For example, with a learned or human-crafted discrete representation of the true state (van den Oord et al., 2017; Hu et al., 2017; Hafner et al., 2021a; Kaiser et al., 2020a) or with a similarity kernel measuring distance between states, Graph Backup might be able to tackle continuous control or tasks with partial observability.

## 8 REPRODUCIBILITY STATEMENT

The code to reproduce all the experiment results are available in the supplementary materials. README includes guides to setup and environment and commands (with hyperparameters) to run the experiments. Experiments for all environments can be run on CPUs but (a single) GPU can speed up for Atari tasks. Besides codes, pseudo-code in Algorithm 2 and Algorithm 1 and hyperparameters described in Appendix D can also be helpful to reproduce the results if readers are interested in implementing by themselves.

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

---

**Algorithm 2** Local Graph Expansion

---

**Input:** source state $S_{\text{source}}$, source action $A_{\text{source}}$, depth limit $d$, breath limit $b$, frequency mapping
    $f : \mathcal{T} \to \mathbb{N}^+$
1: Initialize the set containing states on the boundary of expansion $\mathcal{S}_{\text{new}} \leftarrow \{S_{\text{source}}\}$
2: Initialize the list of expanded state-action pairs $L$, denoting the last element in the list to be $l_{\text{max}}$
3: **for** $i = 0$ to $d$ **do**
4:    Find all transitions on boundary $\mathcal{T}_{\text{new}} \leftarrow \{t | \forall t = (s, a, r, s') \in \mathcal{T}, s \in \mathcal{S}_{\text{new}}\}$
5:    Sample $b$ transitions from $\mathcal{T}_{\text{new}}$ with $p(t) \propto f(t)$, getting $\mathcal{T}_{\text{pruned}} = \{t_1, t_2, ..., t_b\}$
6:    Append state-action pairs to list $L$, $\{l_{\text{max+1}}, l_{\text{max+2}}, ...\} = \{(s, a) | \forall (s, a, r, s') \in \mathcal{T}_{\text{pruned}}\}$
7:    Update boundary states $\mathcal{S}_{\text{new}} = \{s' | \forall (s, a, r, s') \in \mathcal{T}_{\text{pruned}}\}$
8: **end for**
9: **return** $L$

---

## A  OTHER EMPIRICAL FINDINGS

In general, the experiments in three different settings shows Graph Backup consistently brings improvements over multi-step methods. Besides that, we also find that the improvements of $n$-step-$Q$ backup over one-step backup are actually quite limited in the data-efficient setting, whereas Tree Backup performs significantly better than $n$-step-$Q$ backup. This can be explained by the off-policy nature of Tree Backup, as it can bring the benefits of multi-step reward propagation without biasing the value estimation. In data-efficient setting, the flaw of $n$-step-$Q$ is amplified as the learning relies more on historical rather than freshly sampled data. Interestingly, Tree Backup has not received a lot of attention in DRL community. Hernandez-Garcia & Sutton (2019) tested Tree Backup in a toy mountain car experiment which shows $n$-step-$Q$ performs best among multiple multi-step methods, including Tree Backup. (Touati et al., 2018) points out the instability of Tree Backup when combined with functional approximation, both with theoretical analysis and empirically evaluation on some constructed counter-example MDPs. However, our experiments on a larger scale and more diverse set of tasks show Tree Backup has superior sample efficiency when combined with a modern DRL method.

## B  STABLE VALUE ESTIMATE

Graph backup can integrate information from a subgraph, yielding a more accurate and stable value estimation. On the other hand, the nested max operators might lead to overestimation of the value. Here we try to analyse the value estimation given by different backup operators.

We collect 5000 transitions with random walks in a MiniGrid 5x5 Empty Room environment, and have the agents to learn the $q^*$ from this static dataset. In Figure 5, we show the mean and standard deviation of latest 10 estimates of the same state-action pairs. Both mean and standard deviation are averaged over different state-action pairs. The value estimate of Graph Backup quickly stabilize after a few hundred optimization iterations as the mean value converged and the standard deviation reduce to near 0. However, all other backup methods keep giving a varied estimate for the exact same states leading to a higher standard deviation and a fluctuating mean. In terms of over-estimation, the graph backup does give a slightly higher estimate at first (the little bump in the curve), however, it quickly recovers to a stable value.

## C  DATA GRAPH DEFINITION

An MDP data graph is a bipartite multigraph $(\mathcal{S}_{\text{seen}}, \mathcal{N}, \mathcal{E}_{\text{out}}, \mathcal{E}_{\text{in}})$, where $\mathcal{S}_{\text{seen}} \subseteq \mathcal{S}$ is the set of seen state nodes, $\mathcal{N} \subseteq \mathcal{S}_{\text{seen}} \times \mathcal{A}$ is the set of (state conditioned) action nodes, $\mathcal{E}_{\text{out}}$ is a multiset of edges pointing from state nodes to action nodes and $\mathcal{E}_{\text{in}}$ is a multiset of reward-weighted edges pointing from action nodes to state nodes. A state node can point to multiple action nodes because multiple actions might be tried, and the action nodes can point to multiple state nodes because of the stochastic dynamics. When a new transition $(s, a, r, s')$ is observed, edge $(s, (s, a))$ will be added to $\mathcal{E}_{\text{in}}$ and $((s, a), r, s')$ will be added to $\mathcal{E}_{\text{out}}$. A visual example of this data graph can be seen in the Graph Backup diagram in Figure 7(c) with tried (blue) action nodes only.

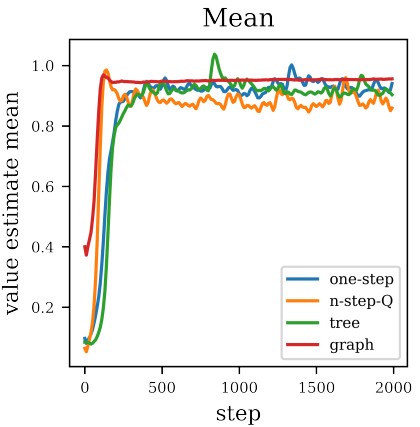 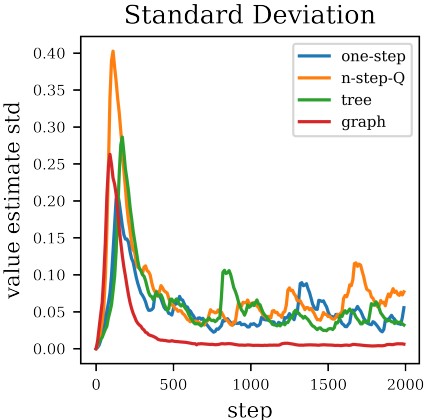

Figure 5: Mean and standard deviation of the value estimate for the same state-action pairs for a fixed dataset collected in MiniGrid Empty Room. The x-axis is the number of optimization steps.

Relating this data graph to Equation (6), we can see $c(s, a)$ is the number of $(s, (s, a))$ edges in $\mathcal{E}_{\text{in}}$ and $f((s, a, r, s'))$ is the number of $((s, a), r, s')$ edges $\mathcal{E}_{\text{out}}$.

## D    DETAILS ABOUT EXPERIMENT SETUP

The Graph Backup and multi-step backup both use a depth limit of 5 for MiniGrid and MinAtar, and 10 for Atari100K. The breath limit for GB-limited is 50 for MiniGrid, 20 for Minatar and 10 for Atari100K.

For MiniGrid and MinAtar, all backup methods are based on the vanilla DQN. The q network has 2 convolutions layers and 2 dense layers, and we follow the hyper-parameters of Rainbow (Hessel et al., 2018) with target network update frequency of 8000, $\epsilon$-greedy exploration with $\epsilon = 0.02$. The learning rate is 0.001 for MiniGrid, 0.000065 for Minatar. The discounting factor is 0.95 for MiniGrid and 0.99 for Minatar. The replay frequency is 1 for MiniGrid, and 4 for Minatar. Since we tested the algorithm in a data-efficient setting, the size of the replay buffer is set to be equal to the overall training steps.

As for Atari100K, our baselines and Graph Backup agents are based on Data-Efficient Rainbow (van Hasselt et al., 2019) with the same hyper-parameters of  Schwarzer et al. (2021).

## E    GRAPH SPARSITY

Across different tasks, we can see a correlation between the density of the transition graph and the improvement of Graph Backup. For MiniGrid tasks where the possible number of states is limited, the Graph Backup brings much larger improvements, whereas, for MinAtar and Atari, the graph is sparse as there are multiple other objects besides the agent that can move in the game world. To analyse the graph density qualitatively, we propose the metric of the novel state ratio. The novel state ratio is the ratio between the number of non-duplicated states and the number of all states that the agent has seen. The novel state ratio will be 1 if there are no overlapping states in the transition graph, in which case Graph Backup reduces to Tree Backup. The average novel state ratio of MiniGrid/MinAtar/Atari are 0.006/0.298/0.927 respectively. The relative average improvements of Graph Backup compared to Tree Backup are 190%/89%/17% on these three group of tasks. The graph density alone, however, is not a reliable indicator to (linearly) predict how much improvement the Graph Backup can bring to a specific task. Although we know the Graph Backup will be the same as Tree Backup if the graph has no crossovers, more crossovers do not always guarantee larger performance improvement.

When we investigate the normalised performance[6] gain and the novel state ratio for each individual task we tested, the correlation is -0.29. Other factors like the structure of the graph and reward density or simply the performance upper bound can also affect the performance gain. As mentioned in Section 4.2, Graph Backup seems to bring more benefits in sparse reward tasks, which can be explained by its counterfactual reward propagation. And the structure pattern of the graph, given the same amount of crossovers, can also play a role.

## F  GRAPH STRUCTURE VISUALISATION

In order to investigate the topology of the graph, we visualize the exact graph structure of the whole transition graph. In Figure 4, we show four representative transition graphs and leave the others to Figure 8 in the Appendix. We apply *radial layouts* as proposed by Wills (1999), which scales well with the number of nodes and aligns well with the transition structure of most of the games. Since the common protocol for evaluating DRL agents in Atari games involves a random number of no-ops before the agent takes over the game, the initial states can vary for different episodes. To adjust for this, we create a hypothetical meta-initial state pointing to all initial states of a game. The meta-initial state is then treated as the root of the whole graph, displayed in the centre of each plot.

A lot of transition graphs of Atari100K show interesting crossover structures that can be leveraged by Graph Backup. For example, the transition graph of *Freeway* forms a windmill-like pattern, where each blade corresponds to a group of trajectories that have connections within the group but between groups. There are also some tasks (e.g. Alien) where crossovers only happen at the start of the game. In such a case, the Graph Backup will not be helpful for most of the source states.

We also find some MinAtar and Atari games have self-loop states (e.g. Asterix), represented as circles in the graph. After further investigations, we found the existence of self-loops is because some of the state transitions in MinAtar and Atari will not make observation changes (such as periodically moving objects). This actually violates the underlining assumptions of Graph Backup that the transitions must be Markovian, which can explain why Graph Backup is inferior to Tree Backup in some of the Tasks. On the other hand, the fact Graph Backup still outperforms multi-step methods on average suggests that Graph Backup is robust against minor violations of Markovian Assumption.

## G  FULL EXPERIMENT RESULTS

In Table 3, we show the results of each individual task and the mean/median of the average performance in each group of tasks.

## H  ALL TRANSITION GRAPHS

In Table 3, we visualise all the transition graphs of Atari100K.

## I  MIXED GRAPH BACKUP

By extending the N-Step-Q backup with the graph structure, we can get another backup target, named *mixed Graph Backup* (GB-mixed). GB-mixed only applies the max operator on the boundary nodes of the transition graph. Define $\mathcal{T}_s \stackrel{\text{def}}{=} \{(\hat{s}, \hat{a}, \hat{r}, \hat{s}') \in \mathcal{T} | \hat{s} = s\}$ and $c(s) = \sum_{T \in \mathcal{T}_s} f(T)$ similarly to before. The GB-mixed target for the state value is then:

$$\bar{G}_s = \begin{cases} \frac{1}{c(s)} \sum_{T \in \mathcal{T}_s} f(T) \left( \hat{r} + \gamma G_{\hat{s}'} \right) & \text{if } c(s) > 0 \\ \max_a q_{\boldsymbol{\theta}'}(s, a) & \text{otherwise} \end{cases} \tag{7}$$

---

[6]The scores of MiniGrid are treated as normalised as it is. The scores of MinAtar are normalised by step 5-million-steps DQN performance reported by Young & Tian (2019)

Table 3: Full Results of all the Tasks. The agent for MiniGrid and MinAtar is based on DQN, and the agent for Atari100K is based on Data-Efficient Rainbow. The default backup operator for rainbow is $n$-step-$Q$. The values in the table for MiniGrid and Minatar are raw game scores and those for Atari100K are human-normalised scores.

| | one-step | $n$-step-$Q$ | tree | graph |
|---|---|---|---|---|
| MiniGrid-LavaCrossingS9N2-v0 | 0.00±0.00 | 0.00±0.00 | 0.00±0.00 | **0.38**±0.21 |
| MiniGrid-Empty-8x8-v0 | 0.69±0.07 | 0.11±0.03 | **0.96**±0.00 | **0.96**±0.00 |
| MiniGrid-SimpleCrossingS9N2-v0 | 0.00±0.00 | 0.00±0.00 | 0.00±0.00 | **0.21**±0.17 |
| MiniGrid-KeyCorridorS3R1-v0 | 0.00±0.00 | 0.00±0.00 | 0.00±0.00 | **0.76**±0.17 |
| MiniGrid-DoorKey-6x6-v0 | 0.00±0.00 | 0.00±0.00 | 0.04±0.04 | **0.58**±0.21 |
| IQM | 0.0 | 0.0 | 0.0 | **0.74** |
| mean | 0.14 | 0.02 | 0.2 | **0.58** |
| median | 0.0 | 0.0 | 0.0 | **0.58** |
| Minatar-seaquest | 0.53±0.11 | 1.12±0.18 | **2.68**±0.17 | 1.77±0.29 |
| Minatar-breakout | 2.42±0.35 | 1.56±0.13 | 4.58±0.14 | **4.95**±0.45 |
| Minatar-asterix | 2.07±0.59 | 2.05±0.40 | **3.33**±0.57 | 2.21±0.53 |
| Minatar-freeway | 0.99±0.56 | 0.29±0.17 | 0.46±0.18 | **30.28**±4.32 |
| Minatar-space_invaders | 9.32±0.57 | 13.78±1.94 | **20.23**±1.30 | **19.93**±1.63 |
| IQM | 1.90 | 1.72 | 3.20 | **8.46** |
| mean | 3.07 | 3.76 | 6.26 | **11.83** |
| median | 2.07 | 1.56 | 3.33 | **4.95** |
| alien | 5.58±0.12 | **9.49**±1.31 | 7.88±0.23 | 6.01±0.56 |
| amidar | 6.49±0.37 | **8.46**±1.10 | 5.06±0.00 | 5.77±0.93 |
| assault | 83.90±1.83 | 42.79±5.25 | 98.27±4.19 | **102.50**±4.28 |
| asterix | 4.74±0.14 | 3.24±0.59 | **7.11**±1.61 | 3.50±0.38 |
| bank_heist | 2.72±0.64 | 8.26±1.04 | **63.31**±19.18 | 38.13±20.33 |
| battle_zone | 9.31±2.53 | 21.83±1.09 | 22.64±3.80 | **29.51**±4.62 |
| boxing | -98.42±40.41 | -16.63±4.91 | **77.19**±35.70 | 53.14±45.08 |
| breakout | 48.73±10.02 | 30.53±1.23 | 57.30±4.30 | **65.30**±7.05 |
| chopper_command | 8.52±1.15 | 9.50±1.25 | 6.36±2.13 | **11.08**±1.50 |
| crazy_climber | 101.27±27.40 | 34.85±4.54 | 110.68±8.89 | **113.45**±17.38 |
| demon_attack | 24.65±6.86 | 3.40±0.68 | **24.84**±4.00 | 23.83±5.59 |
| freeway | 53.29±17.20 | **61.25**±22.38 | 35.81±22.65 | 64.16±20.67 |
| frostbite | 3.84±0.14 | 4.56±0.14 | 4.31±0.07 | 4.40±0.12 |
| gopher | **18.48**±5.57 | 7.40±1.14 | **19.48**±4.91 | 16.54±5.34 |
| hero | 17.08±6.95 | 22.54±1.18 | **31.25**±3.28 | **30.63**±1.23 |
| jamesbond | 75.72±25.42 | 53.47±6.90 | **92.52**±7.21 | 78.80±10.82 |
| kangaroo | 29.61±4.05 | 143.46±40.42 | 76.86±27.87 | **141.14**±37.87 |
| krull | 154.52±19.33 | 112.99±16.28 | 159.09±6.34 | **174.77**±17.05 |
| kung_fu_master | **93.31**±19.35 | 28.07±4.81 | 62.37±4.70 | **92.06**±10.15 |
| ms_pacman | 9.71±0.38 | 12.94±1.31 | 9.11±1.06 | **14.60**±1.11 |
| pong | 1.15±0.35 | **70.69**±14.10 | 11.10±6.90 | 27.00±2.43 |
| private_eye | 0.11±0.00 | 0.11±0.00 | 0.11±0.00 | -0.05±0.00 |
| qbert | 3.97±0.59 | 8.75±2.97 | 13.41±2.85 | **19.27**±3.69 |
| road_runner | **167.53**±63.50 | 42.90±9.81 | 116.12±51.72 | **164.74**±58.58 |
| seaquest | 1.51±0.12 | 0.96±0.11 | 1.14±0.18 | 1.06±0.19 |
| up_n_down | 25.33±1.72 | 20.84±0.73 | 20.76±2.95 | **31.45**±1.32 |
| IQM | 18.85 | 16.55 | 26.76 | **34.25** |
| mean | 32.8 | 28.72 | 43.62 | **50.49** |
| median | 13.39 | 16.89 | 23.74 | **30.07** |

Table 4: The unnormalised scores for Atari100K. Note that taking average of these scores will lead to a evaluation metric highly weighted by games with higher score range.

| | one-step | n-step-Q | tree | graph |
|---|---|---|---|---|
| alien | 612.93±8.14 | 714.60±88.06 | 771.30±15.97 | 642.70±38.67 |
| amidar | 117.01±6.37 | 179.38±3.76 | 92.51±0.00 | 104.65±16.04 |
| assault | 658.32±9.51 | 422.47±16.69 | 733.00±21.74 | 754.98±22.23 |
| asterix | 603.33±25.95 | 497.25±8.16 | 799.67±133.22 | 500.67±31.82 |
| bank_heist | 34.27±4.74 | 79.50±1.24 | 482.00±141.69 | 295.97±150.20 |
| battle_zone | 5603.33±880.01 | 11800.00±0.00 | 10243.33±1323.66 | 12636.67±1608.85 |
| boxing | -11.71±4.85 | 0.00±0.00 | 9.36±4.28 | 6.48±5.41 |
| breakout | 15.73±2.88 | 10.60±0.00 | 18.20±1.24 | 20.51±2.03 |
| chopper_command | 1371.67±75.49 | 1103.00±0.00 | 1229.33±140.01 | 1539.67±98.61 |
| crazy_climber | 36148.17±6862.90 | 22990.00±0.00 | 38503.67±2226.05 | 39198.67±4354.52 |
| demon_attack | 600.42±124.86 | 182.00±11.81 | 603.90±72.76 | 585.48±101.56 |
| freeway | 15.78±5.09 | 14.96±6.69 | 10.60±6.70 | 18.99±6.12 |
| frostbite | 229.07±13.21 | 246.15±6.46 | 249.37±3.01 | 252.93±5.05 |
| gopher | 655.77±120.05 | 341.50±11.23 | 677.40±105.83 | 613.93±115.14 |
| hero | 6116.58±2072.96 | 7414.70±0.00 | 10340.43±978.17 | 10154.15±365.49 |
| jamesbond | 236.33±69.60 | 140.75±7.04 | 282.33±19.74 | 244.75±29.63 |
| kangaroo | 935.33±120.86 | 1907.00±207.06 | 2344.67±831.42 | 4262.33±1129.61 |
| krull | 3247.53±206.32 | 3004.35±13.30 | 3296.30±67.65 | 3463.70±182.04 |
| kung_fu_master | 21233.33±4348.28 | 9761.50±794.92 | 14277.67±1055.47 | 20952.00±2281.75 |
| ms_pacman | 952.40±25.32 | 1015.65±6.69 | 912.40±70.82 | 1277.30±73.99 |
| pong | -20.29±0.13 | 0.00±0.00 | -16.78±2.44 | -11.17±0.86 |
| private_eye | 100.00±0.00 | 0.00±0.00 | 100.00±0.00 | -10.34±0.00 |
| qbert | 692.00±78.34 | 1176.75±0.00 | 1945.67±379.41 | 2724.75±489.89 |
| road_runner | 13134.67±4973.98 | 4187.00±0.00 | 9108.00±4051.97 | 12916.67±4588.43 |
| seaquest | 703.13±50.67 | 529.20±0.00 | 596.27±69.83 | 514.53±78.49 |
| up_n_down | 3360.67±191.43 | 3303.50±0.00 | 2849.73±329.41 | 4043.07±147.85 |
| mean | 3744.07 | 2731.61 | 3863.86 | 4527.08 |

The GB-mixed target for the state-action value is then a frequency-weighted average of the next state target:

$$\bar{G}_s^a = \begin{cases} \frac{1}{c(s,a)} \sum_{T \in \mathcal{T}_{s,a}} f(T) \left( \hat{r} + \gamma G_{\hat{s}'} \right) & \text{if } c(s,a) > 0 \\ q_{\theta'}(s,a) & \text{otherwise} \end{cases} \tag{8}$$

Similar to N-Step-Q backup, GB-mixed is not a strictly off-policy backup operator. The value of boundary states is estimated in an off-policy manner while the rewards of interior paths are on-policy, hence the name *mixed* Graph Backup. GB-mixed is a biased backup operator when evaluating the target policy. However, it can also be less noisy than GB-nested since there the nested max operators can propagate over-optimistic value estimation error from every step to the source state.

## J    GRAPH DENSITY OF ATARI100K

We also explore the role of graph density in the same set of tasks. In Figure 6, we show the correlation between relative performance and novel state ratio, where each point represents a task in Atari100K. The relative performance is the human normalised score of GB-nested divided by that of Tree Backup. It indicates how many benefits the agent can get from leveraging the graph structure. There is a negative correlation of -0.22 between the novel states ratio and relative performance. Although the correlation is weaker than in the case of cross-group comparison, the graph density can still affect the effectiveness of Graph Backup.

Figure 6: Relative performance and novel state ratio

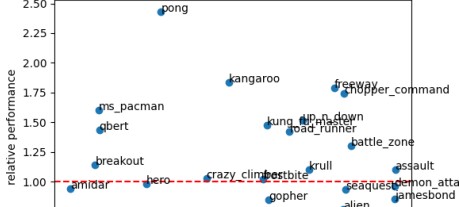

## K    MINATAR 1M

Table 2 shows the performance of different methods after 1M steps of training. Both one-step and Graph Backup achieves the means scores close to the DQN asymptotic performance reported by Young & Tian (2019). This shows that even with more training data, Graph Backup is able to converge to the same level of performance as one-step backup. Surprisingly, though, the $n$-step-$Q$ backup and Tree Backup both perform worse than one-step backup with more data. This can be explained by the innate problems of $n$-step-$Q$ and nested max operators. Strictly speaking, $n$-step-$Q$ is not an off-policy backup operator because it always uses online reward sequences for the estimation of its value, which can counter its advantages in longer reward propagation. This can be especially true for Minatar because its framerate is much lower than vanilla Atari and longer reward propagation may not be as important. For Tree Backup and Graph Backup, the nested max operator may cause an overestimation of values. We leave methods that address this problem for future work. With the over-estimation dealt with properly, Graph Backup may show even stronger performance in an asymptotic setting.

## L    COMPUTATIONAL OVERHEAD

In contrast to model-based planning methods, Graph Backup does not bring computational overhead at action selection. Therefore, during deployment, decision latency and the computational cost will be the same as for previous methods, which we believe is important in a Data-Efficient setting, where it's both important and required to leverage additional computing at training time to make up for the limited number of samples from the environment. On the other hand, the computational overhead of Graph Backup during training is nuanced and is heavily dependent on the implementation, the base algorithm, hyperparameters and the network architecture. In our implementation, the subgraph is constructed from the adjacency list (python dictionary), and the value for expanded state nodes can be computed in a batched way. This is quite different from model-based planning methods like MCTS where the tree expansion is conditioned on the policy and value estimation of parent nodes. Due to the easy parallelization of the neural network inference, the main bottleneck becomes the sub-graph expansion itself. Our implementation did the graph expansion and backup with python code and the different samples in a batch are processed sequentially, which makes the training 2-5 times slower than a one-step backup. However, this is just a simple proof of the algorithm's performance, and the implementation could easily be sped up drastically by JIT compilation or multi-threading (or multi-processing).

## M    VALUE ESTIMATION VISUALIZATIONS

In Figure 7, we visualize the value estimates for all positions in an empty room environment. An agent starts at the bottom left corner and tries to reach the top right goal position. We fix the value of the goal position to be 1, and all other value estimations come from the Q network when the agent is in such a position and facing right. The discount factor $\gamma = 0.95$ is the same as in our empirical evaluations so the ground truth value for the initial position is $0.57$. We fixed the random seeds of the agents and all of them will find the first reward in step 1957 and the training starts at step 2000. We can see graph backup quickly converge to the ground truth value estimation for half of the states at step 3000, whereas other backup methods only failed to correctly estimate the value of the initial state even after 30000 steps of training.

An interesting observation is the value is not overestimated for Graph Backup even though it has a lot of nested max operators. On the other hand, values of some states are over-estimated by Tree Backup agent and one-step backup agents at step 30000. This can be explained by the combination of counterfactual credit assignments and online interactions. If the value of a state is overestimated, such excess value will quickly propagate to a lot of the interconnected states. Therefore, the agent will then actively try to reach the over-estimated states and such overestimation will then be corrected. Another interpretation is the overestimation comes from the extra noise of one-step and Tree Backup. In Figure 5, even training the value from the logged data, Tree Backup and one-step backup will sometimes give higher value estimation than Graph Backup.

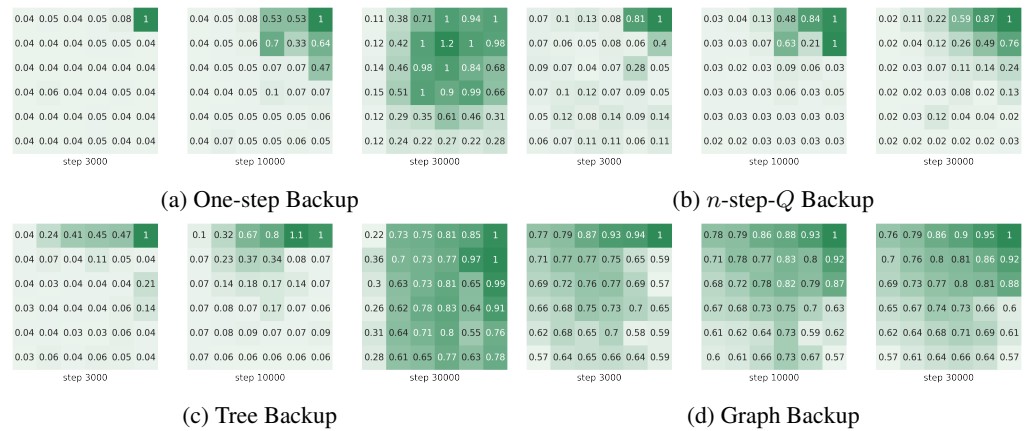

Figure 7: Value Map of Empty Room

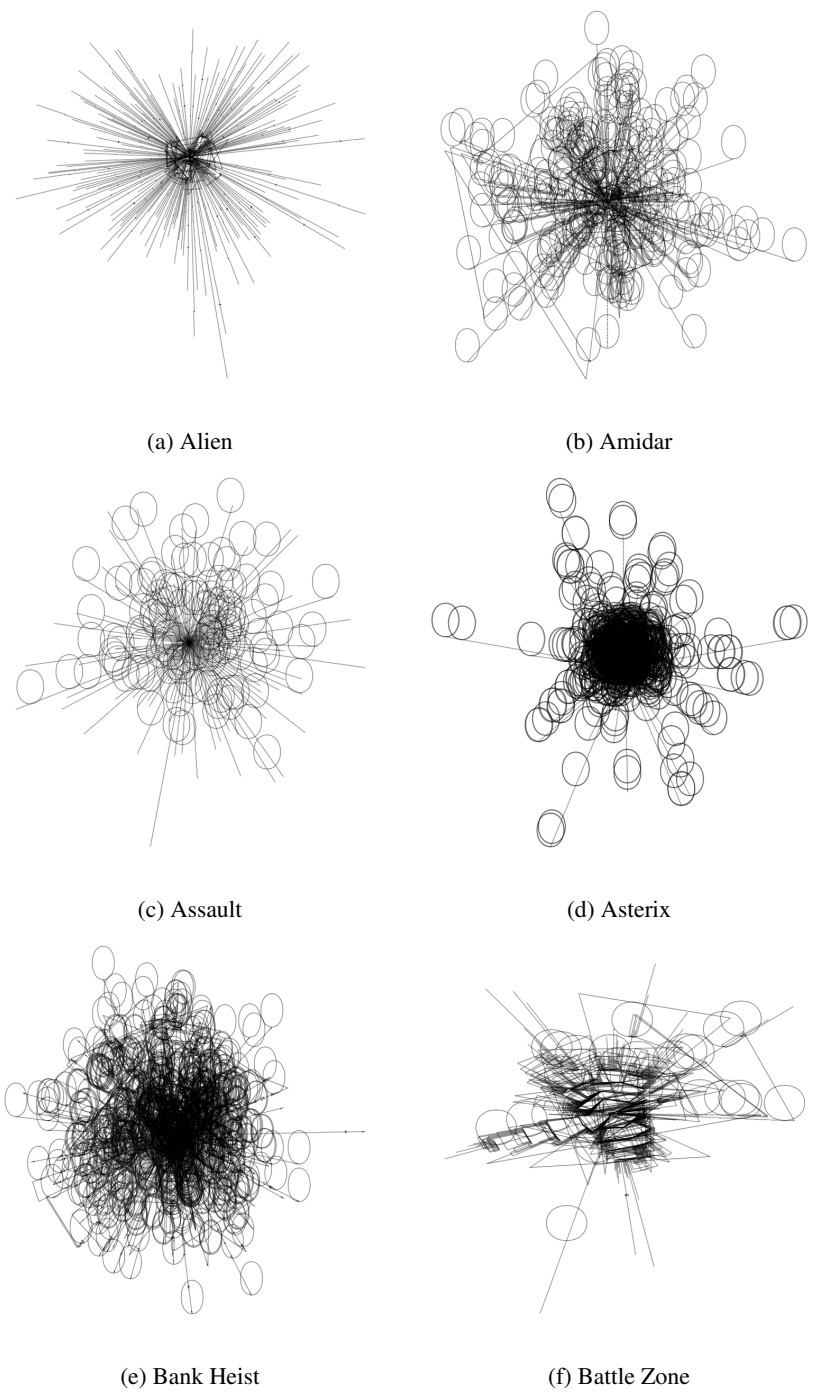

(a) Alien

(b) Amidar

(c) Assault

(d) Asterix

(e) Bank Heist

(f) Battle Zone

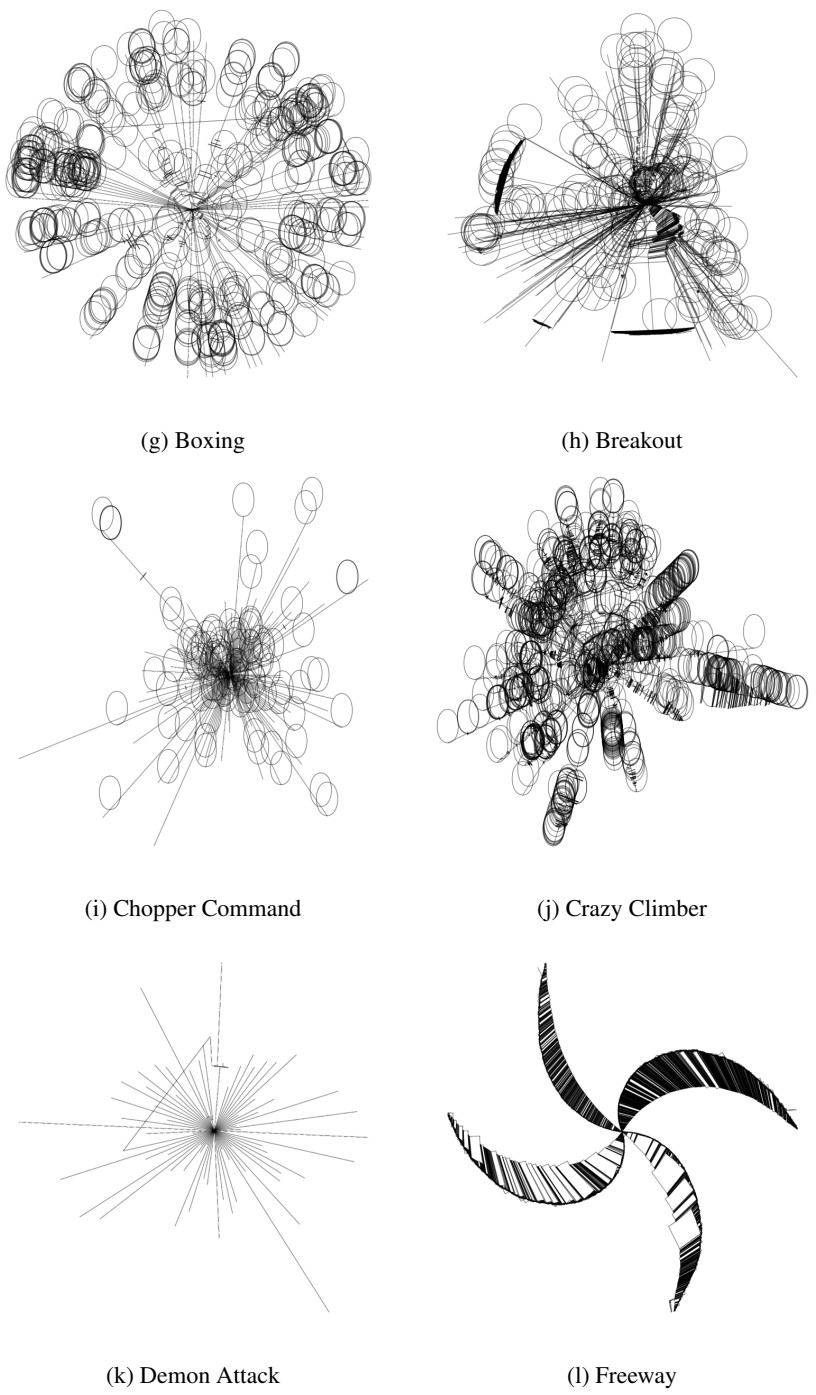

(g) Boxing

(h) Breakout

(i) Chopper Command

(j) Crazy Climber

(k) Demon Attack

(l) Freeway

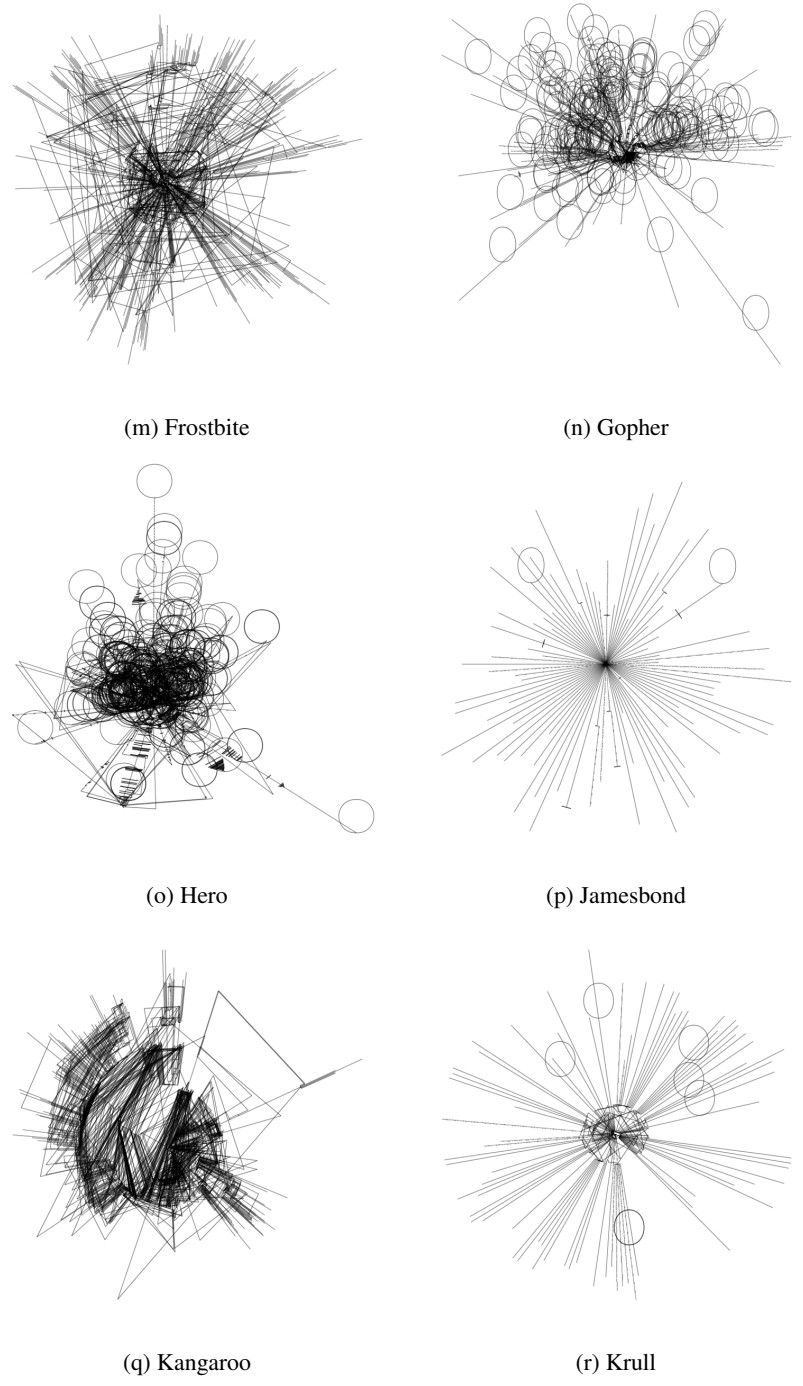

(m) Frostbite

(n) Gopher

(o) Hero

(p) Jamesbond

(q) Kangaroo

(r) Krull

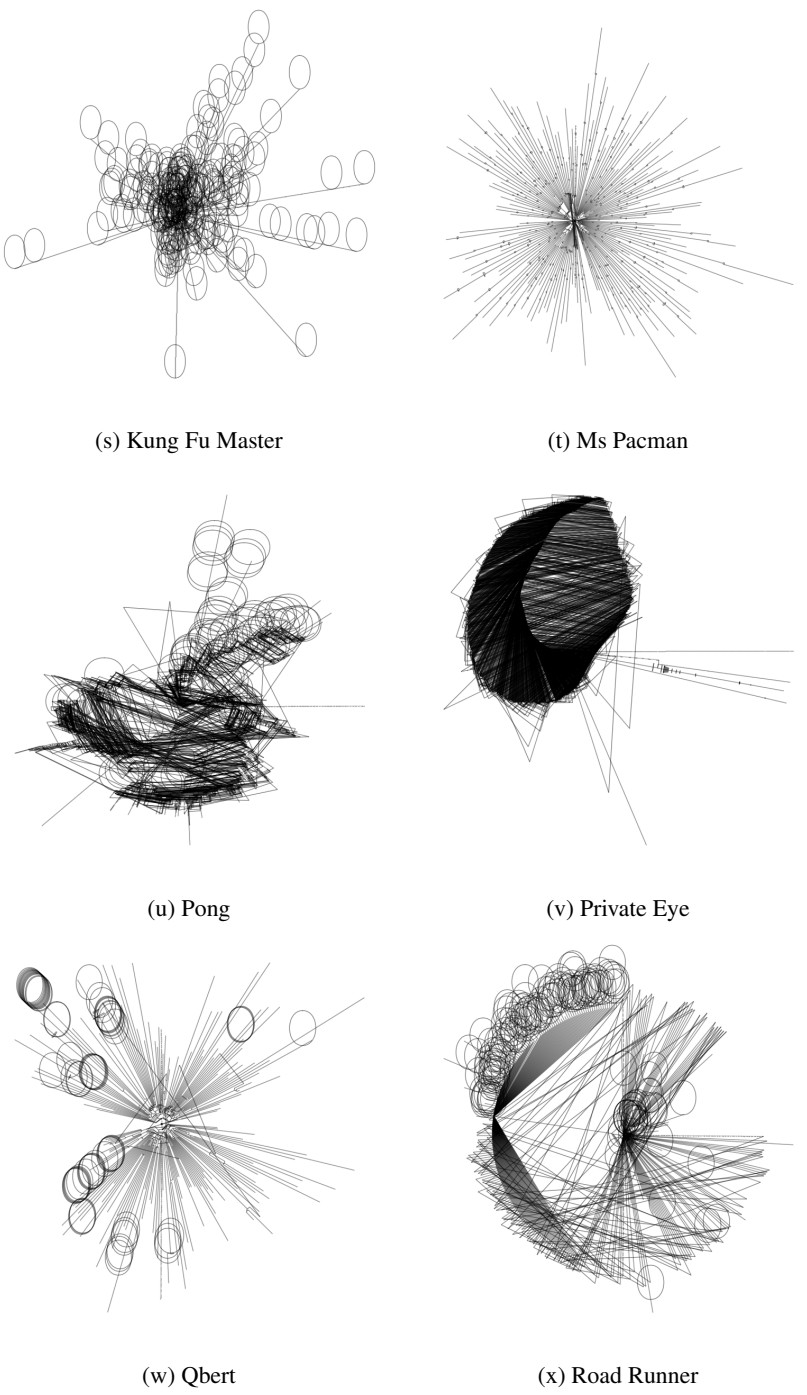

(s) Kung Fu Master

(t) Ms Pacman

(u) Pong

(v) Private Eye

(w) Qbert

(x) Road Runner

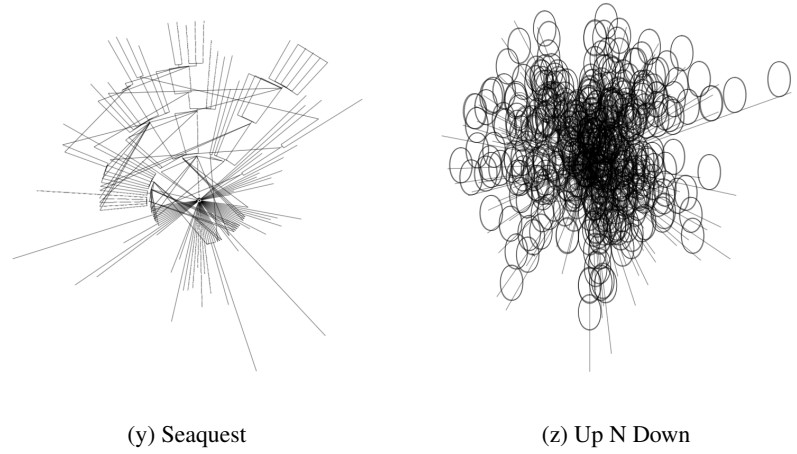

(y) Seaquest         (z) Up N Down

Figure 8: All Transition Graphs of Atari100K

