# OpenReview forum: "Graph Backup: Data Efficient Backup Exploiting Markovian Transitions"
_ICLR.cc/2023/Conference — Submitted to ICLR 2023_

### Official Review · Reviewer_iGcV · 2022-10-23

**Confidence:** 4
**Correctness:** 3
**Technical Novelty And Significance:** 3
**Empirical Novelty And Significance:** 3
**Recommendation:** 5

**Clarity, Quality, Novelty And Reproducibility:**

# Detailed Comments (Clarity, Quality, Novetly, Reproducibility)

-   Section 1, (data-use): "Data is normally stored in a buffer and only
    used several times for learning before being discarded. This means
    these algorithms underutilise the data that is generated from the
    simulator"

    It is hard to argue that replay-buffer based training doesn't make
    use of data. As far as minimizing TD errors of the current policy,
    which is the goal of most value-based methods, the data has
    contributed all it can in some sense. While the environment may
    possess additional structure, this is not known apriori. Of course,
    it can also be argued that the right back-up operator can minimize
    these errors in fewer iterations.

-   Section 2, (Connection to Model-based RL): Paragraph 3 lists some
    model-based work and separates the proposed graph backup from MCTS
    methods that also utilise tree-structure search algorithms. It is
    would be good to discuss further what advantages graph backup
    provides over background planning methods, that use a learned model
    to provide samples. This would bring all problems - including those
    without a simulator - closer to "cheap data" regime.

-   Section 3, (Equation 1): It is not quite correct to refer to this as
    a loss function. This is because temporal difference methods which
    use a stop-gradient are not taking the gradient of any objective
    function. There are gradient TD methods, but they minimize the
    projected mean squared bellman error.

-   Section 4 (counter function) : This seems to be the key function in
    graph backup - but it is also the most problematic. Similar to
    count-based exploration methods, accurately learning to count
    visitation is a challenging problem.

-   Section 4.2 (counterfactual): A tabular update to the value function
    at one transition will not directly influence the value function at
    other states from other trajectories. With function approximation,
    however, there will be generalization which may allow for changes in
    nearby states in feature space.

-   Section 4.2 (spare reward settings): if there are truly no rewards
    from a lack of successful trajectories, then both graph backup and
    one-step backups will fail to learn anything meaningful. If there
    are some trajectories that are successful, then this information
    will still be propagated to other states from other trajectories
    through generalization, as long as they are in the replay buffer. It
    may take more iterations but this is not what you seem to argue in
    Section 4.2 Paragraph 1.

-   Section 4.3 (variance): this point about averaging over many
    trajectories is interesting and valid. It is, in my opinion however,
    lacking some nuance because it may not be the case that your counter
    functions is accurate.

-   Section 4.2 (bias): while you discuss how graph backup can reduce
    variance, you also mention that multi-step backups can be biased for
    Q-learning. I do not see a discussion of whether graph backup is
    similarly biased. This would help situate graph-backup in the space
    of backup operators described in section 3.

-   Section 4.3 (Limiting expansions in Figure 1b). Could you explain
    why the top node in figure 1b is faded? Its parent node only expands
    one child node, so the top most node should satisfy the breadth
    limit, unless the breadth is not at the expansion level but at the
    tree level.

-   Section 4.5 (crossover assumption): I think it would benefit more
    broadly to discuss (or demonstrate empirically) the importance of
    these assumptions. The Markov assumption is obviously tied to
    effectiveness of not only the method but RL algorithms as whole. It
    is not, however, obvious the degree that crossovers matter (for
    example, if 1\% vs 10\% of states have a crossover).

**Strength And Weaknesses:**

# Strengths

-   The empirical results presented seem strongly in favour of graph
    backup. This is especially interesting because a priori I would not
    expect that the results would be strong on Atari due to low expected
    crossover.

-   Overall easy to follow with a clear contribution. Graph backup is
    well-connected to the literature, as a generalization in breadth
    from tree-backup. Experiments also leverage an already strong
    baseline, and tweak a single component making the contribution of
    graph backup clear as well.

# Weaknesses

-   Despite the paper being overall clear and easy-to-follow, there are
    some statements in the paper that are unclear. While I understand
    the method, its motivation and its use in Rainbow DQN, I am not able
    to follow the reasons for the large improvement. The difference
    between tree backup and graph backup, in particular, is
    significantly larger than I would expect. Little intuition, or
    empirical insight, is given to explain this effectiveness. Although
    the crossover rate for Atari is reported, I am still left wondering
    why the method is as performant as reported.



**Summary Of The Paper:**

This paper proposes a new backup operator that exploits potential graph
structure in the observed transitions. The purpose of this backup
operator is to increase data efficiency by allowing information to be
backed-up further along state transitions and providing better value
estimates. Their results demonstrate, rather conclusively in the
aggregated score, that backup operators can improve data-efficiency. In
particular, a rainbow DQN agent augmented with graph backup tends to
outperform one-step, n-step and tree-backup targets on MinAtar and Atari
100k.


**Summary Of The Review:**


I am somewhat torn on this submission. On the one hand, the contribution
is clear and the experiments are in favor of the proposed method (graph
backup). It is also well connected to previous literature as an
extension of tree-backup. This suggests that graph-backup does improve
data-efficiency, and the main explanation is that this improvement is
due to exploiting the graph structure of some fixed breadth. Some
statements in the manuscript are still unclear, and the overall
motivation for why graph backup would benefit in environments like Atari
is still non-obvious. In light of the large improvements that graph
backup can make, I think this finding needs to be better explained,
either from further intuition or results on a simple problem (but with
function approximation). I am currently rating this paper at a weak
reject. However, I am very open to increasing my score up to an accept.

---

> ### Author Response · Authors · 2022-11-17
> **Individual Response (1)**
>
> Thanks for your compliments on the work, and for your feedback and questions, we think that addressing them has improved the clarity and correctness of the work a lot.
>
> > overall motivation for why graph backup would benefit … this finding needs to be better explained, either from further intuition or results on a simple problem (but with function approximation).
>
> Following your suggestion, we add a new analysis about the value estimation of DQN on EmptyRoom with backup methods in Appendix M Figure 7. We also discussed why Graph Backup is helpful in the General response, in particular giving intuition for why it improves over other methods even on Atari.
>
> > "Data is normally stored in a buffer … algorithms underutilise the data that is generated from the simulator" It is hard to argue that replay-buffer based training doesn't make use of data. As far as minimizing TD errors of the current policy, which is the goal of most value-based methods, the data has contributed all it can in some sense.
>
> Thanks for the suggestion. We’ve removed such an argument and instead focus on the graph structure vs. chain structure.
>
> >  It is not quite correct to refer to this as a loss function. This is because temporal difference methods which use a stop-gradient are not taking the gradient of any objective function.
>
> We are not sure if we understand this particular point. Eq 1 is called the loss function starting from [1] and $\\theta$ indeed takes gradient from it. Stop gradient only happens to $G$, which can be regarded as a label.
>
> [1] Playing Atari with Deep Reinforcement Learning, 2013
>
> > This seems to be the key function in graph backup - but it is also the most problematic. Similar to count-based exploration methods, accurately learning to count visitation is a challenging problem.
>
> To clarify, this counter function plays two roles 1) the adjacency list of the graph; 2) to weight transitions when the same action leads to different future states. The first role is much more important. To make Graph Backup effective, there is a requirement for the data to have crossovers between trajectories, which means it only needs a state in the whole trajectory to be a revisited state. This is a weaker requirement than count-based exploration methods, which need most of the states to be visited (or pseudo-visited) multiple times. We acknowledge that accurately counting visitations is a challenging problem, but our results show that even when you might not expect it (i.e. in Atari games), Graph Backup can provide benefits even without a more involved way of counting visitations than exact state matching
>
> > Section 4.2 (counterfactual): A tabular update to the value function at one transition will not directly influence the value function at other states from other trajectories. With function approximation, however, there will be generalization which may allow for changes in nearby states in feature space.
> Section 4.2 (spare reward settings): …. If there are some trajectories that are successful, then this information will still be propagated to other states from other trajectories through generalization...
>
> In section 4.2, we are talking about counterfactual reward propagation in a single backup. We agree generalization of the function approximator and multiple iterations of backup might bring some similar effects and eventually backpropagate the reward counterfactually, but they will also bring troubles like the deadly triad [2]. Too many iterations of DQN training without enough online interactions coming in will often cause training to diverge, which we also observed in MinAtar experiments.
>
> [2] Deep Reinforcement Learning and the Deadly Triad, 2018
>
> > this point about averaging over many trajectories is interesting and valid. It is, in my opinion however, lacking some nuance because it may not be the case that your counter functions is accurate.
>
> For the variance reduction, we are claiming something smaller than averaging over trajectories. Basically, for a single state $s$, the backup shouldn’t give varied value estimation given a fixed replay buffer. While this should be obvious, one-step backup and multi-step backup cannot guarantee this. This is shown empirically in Figure 5 in the Appendix.
>
> > while you discuss how graph backup can reduce variance, you also mention that multi-step backups can be biased for Q-learning. I do not see a discussion of whether graph backup is similarly biased. This would help situate graph-backup in the space of backup operators described in section 3.
>
> We claimed $n$-step-$Q$ is biased but not for other multi-step methods. Tree backup, for example, is unbiased. Please see [3] for details. We similarly believe that Graph Backup is unbiased, which could be proven in the same way as Tree Backup, but we haven’t proven it formally.
>
> [3] Understanding Multi-Step Deep Reinforcement Learning: A Systematic Study of the DQN Target, 2019

---

> > ### Author Response · Authors · 2022-11-17
> > **Individual response (2)**
> >
> > > Section 4.3 (Limiting expansions in Figure 1b). Could you explain why the top node in figure 1b is faded?
> >
> > That’s because the breath limit counts transitions (edges with blue squares) and not state nodes. Given we already have two transitions at this depth, we don’t recurse onto the third one. Such a design is to make sure the computation time and memory of Graph Backup are bounded since otherwise, a densely connected graph can have a huge amount of transitions. In effect, faded nodes are ones not visited by Graph Backup with the set depth and breadth hyperparameters, but could be visited if the breadth (in this case) was increased.
> >
> > > I think it would benefit more broadly to discuss (or demonstrate empirically) the importance of these assumptions. The Markov assumption is obviously tied to effectiveness of not only the method but RL algorithms as whole. It is not, however, obvious the degree that crossovers matter (for example, if 1% vs 10% of states have a crossover)
> >
> > Yes, we agree it will be interesting if we can find some correlation between the novel state ratio and performance gain of Graph Backup. In Figure 6 in the Appendix, we try to plot the novel state ratio and performance gain of each single Atari game. In general, having more crossovers are helpful but it turns out the correlation is rather weak. We believe that means the effectiveness of Graph Backup is also affected by the exact graph structure and other properties of the MDP.
> >
> >
> > Thanks again for your feedback. We hope that we’ve addressed your questions and suggestions regarding clarity, and we hope that we’ve been able to provide better intuition and justification for why Graph Backup performs as well as it does on Atari. If you have other criticisms or suggestions then please let us know, and if not we’d appreciate you raising your score.

---

> > > ### Comment · Reviewer_iGcV · 2022-12-01
> > > **Thank you for addressing most of my concerns, but points raised by other reviewers have impacted my decision as well.**
> > >
> > > Thanks for addressing most of my concerns. I like the paper, and quite sympathetic to its goals. After reading the other reviews, however, I am not as open to increasing my score as in my original review.
> > >
> > > Some relevant issues were brought up in other reviews that I feel should be addressed in a re-submission.
> > >
> > > - Atari benchmark: I do understand that there is a low seed count culture in deep RL, especially in Atari. The Atari100k and minatar benchmarks are less demanding, and could realistically be run on a larger number of seeds. The other, and perhaps arguably more pertinent issue in my decision, is the realization that no randomization (or noise) was added in the start-state such as no-op starts. I believe this is standard practice, but no details are provided for the wrapper setting used.
> > >
> > > - I also did not consider the scalability of the method. I think scalibility alone is not an issue, but it may be relevant if it is an order of magnitude less efficient. It would be enlightening to benchmark the wallclock time of the overall algorithm using graph backup (including graph creation and value estimation).
> > >
> > > - The results in Appendix M are interesting in that they clearly demonstrate the improvements in propagating value. But, many things are unclear. This is not tabular, so what are the features? Then, what algorithm is used? I am rather surprised that 30k steps is not sufficient to estimate the value or return the optimal policy. Is this a continuing environment?
> > > This also does not directly translate to improvements in control, which is the main demonstrated benefit of using graph backup. They are related, of course, but it is difficult to interpret Figure 7 in light of this because only state values are reported.

---

> > > > ### Author Response · Authors · 2022-12-02
> > > > **Thanks and Further Clarification**
> > > >
> > > > Thanks for your further input! Here's the response to some of the new concerns you raise.
> > > >
> > > > > The other, and perhaps arguably more pertinent issue in my decision, is the realization that no randomization (or noise) was added in the start-state such as no-op starts. I believe this is standard practice, but no details are provided for the wrapper setting used.
> > > >
> > > > We agreed this is the standard treatment and we **do** incorporate that into that our Atari experiments because all of our setups are the same as Data Efficient Rainbow. Please see graphbackup/atari/src/rlpyt_atari_env.py L75 and L135 for details.
> > > > To clarify, the transition graph looks to start from the same position because we add a hypothetical initial state when plotting the graph for better visualization. But the transition graph seen by the graph backup is do not have this hypothetical initial state.
> > > >
> > > > > The results in Appendix M are interesting in that they clearly demonstrate the improvements in propagating value. But, many things are unclear.
> > > >
> > > > The setting in Appendix M is exactly the same as our MiniGrid experiments. So that's online learning progress with both data-collection (exploration) and value learning, which is different from the setting in Appendix B Figure 5 (offline learning)
> > > >
> > > > > This is not tabular, so what are the features?
> > > >
> > > > Inputs are the feature map (symbolic pixels), which is exactly the same as all of our MiniGrid experiments.
> > > >
> > > > >  Then, what algorithm is used?
> > > >
> > > > That's an online DQN.
> > > >
> > > > > I am rather surprised that 30k steps is not sufficient to estimate the value or return the optimal policy. Is this a continuing environment?
> > > >
> > > > We believe that’s because rewards are quite sparse for the transitions graph. The agent should first receive the first reward with random walks, which takes 2000 steps for the random seed we used to plot Figure 7. And then the rewards can only be propagated correctly if the correct one of these 2000 transitions is sampled in further updates (for one-step method). These luck samplings will have to happen multiple times until the reward is propagated to the initial state.
> > > >
> > > > > This also does not directly translate to improvements in control, which is the main demonstrated benefit of using graph backup. They are related, of course, but it is difficult to interpret Figure 7 in light of this because only state values are reported.
> > > >
> > > > Thanks for the suggestion, we’ll try to visualize optimal paths according to the q-values in the next version. For now, the control performance after 100K iterations can be found in the second row of Table 3. For the particular seed used to generate Figure 7. The one-step backup started to get positive rewards in evaluate mode after 67k steps of training; that of n-step-Q is 59k steps, that of tree backup is 31k steps, and that of graph backup is 7k steps.
> > > >
> > > > > I also did not consider the scalability of the method. I think scalability alone is not an issue, but it may be relevant if it is an order of magnitude less efficient. It would be enlightening to benchmark the wallclock time of the overall algorithm using graph backup (including graph creation and value estimation).
> > > >
> > > > In the revised paper, we've added a new section in Appendix L to discuss the computational cost. While we agree to benchmark this in more detail, we argue this is not a major reason to reject the paper because:
> > > >
> > > > 1. the computational cost is highly dependent on the implementation but there is no fundamental reason that Graph Backup will be an order of magnitude less efficient.
> > > > 2. The computational cost is not the main concern for data-efficient methods.
> > > > 3. In our non-parallel proof-of-concept implementation, Graph Backup is 2-5 times slower than the single-step method, depending on the graph sparsity, and we don't think this is a major problem.

---

### Official Review · Reviewer_zkNq · 2022-10-24

**Confidence:** 4
**Correctness:** 4
**Technical Novelty And Significance:** 3
**Empirical Novelty And Significance:** Not applicable
**Recommendation:** 5

**Clarity, Quality, Novelty And Reproducibility:**

I use 4 levels of grades: strong, good, fair, poor

Clarity: poor. See point (5) of weaknesses.
Quality: fair.
Novelty: good.
Reproducibility: good.

**Strength And Weaknesses:**

Strengths:
1) The problem being studied is well-motivated: data efficient RL is a practical and important topic in many real-world RL tasks.
2) The idea of using crossover trajectories in updating the value function is interesting and novel. I am a bit surprised that this has not been studies in previous works (from what I saw in the discussion of related works).
3) The empirical results look promising.
4) The analysis in Section 6 is an addition. Though there might be some limitations there.

Weaknesses:
1) The settings with overlapped (crossover) trajectories are often quite restricted. In the Atari game, the ratio of crossover trajectories is relatively high because the initial state is always the same. This is a strong assumption for the majority of the RL tasks/MDPs. Therefore, this has strongly limited the applicability of the proposed Graph Backup method.
2) It is not clear why a data graph is necessary for the proposed backup in Equation (6), and it looks like that this critical information is only included in the Appendix. Therefore it is very hard to evaluate the motivation of the proposed data graph.
3) In computing the new backup, there is a computational overhead since it requires more computations with the iterative update and multiple crossover trajectories. It would be good to add some discussions/evaluations of the computational overhead.
4) Section 4.2: This is more like a discussion. While it is good to have such kind of discussions, the paper would be stronger if it includes some formal results for the described advantages.
5) The writing of the paper needs significant work. While this is normally not the reason to reject a paper, it is making it really challenging to evaluate the paper itself when clarify is a serious issue. Examples:
- Figure 2 is far from where it is first referred to. It does not really help understand why multi-step backup adds variance to the state value estimates.
- The visualization of Figure 1(a) is good, but it would be better if there is explanation as to what the shape of the graph indicates, e.g., is it a subgraph that shares the same target state? It will be more informative if such kind of discussion/description is provided.
- In Figure 1, "The blue squares represent the state-action pairs" -> they should just be the actions from what I understand
- Equations (7) and (8) are in the appendix, but they are referred to in the main text.
- Some typos and grammar issues. E.g., 2nd last line of Page 2: "as they are also utilise" -> remove "are"; 1st line of Page 3: "Zhu et al. (2020) propose the using the MDP graph" -> "propose to use"; 3rd line after Equation (3): "in a off-policy" -> "an"; 4th last line of Page 3, "Despite what it’s name suggests"->"its". There are a few more which I did not list but strongly suggest the authors check them out.
- Notations:
1st line after Equation (1): $G^{a_T}$ should be $G^{a_t}$; Equation (2):
$\theta'$ is not explained? You did not mention anything about a separate network. Also, the superscript of G is not explained, and is inconsistent with Equations (3) (4);
Equation (4): why do we compare t with n in the condition  of Equation (4)? by t<n do you mean the last time step in n-step return? In this case you should use a t' as the generic step, and the condition should be $t'<t+n$.


**Summary Of The Paper:**

The paper studies the value function back-up estimation problem in data efficient reinforcement learning. The key idea is to the Graph Backup, with uses creates a tree-structured update of the value function based on multiple trajectories which have overlapping states. Because of the overlapping trajectories, it has advantages such as the counterfactual credit assignment and smaller variance. The empirical results show that the proposed Graph Backup method, when integrated into DQN or Rainbow, outperforms existing backup methods in data efficient settings.

**Summary Of The Review:**

Main reasons of accepting the paper are points (1)(2)(3) of strengths.
Main reasons of rejecting the paper are points (1)(2)(5) of weaknesses.

---

> ### Author Response · Authors · 2022-11-17
> **Individual Response**
>
> Thanks for all your suggestions and comments, we think the paper is improved by them. To address your comments:
>
> > The settings with overlapped (crossover) trajectories are often quite restricted. In the Atari game, the ratio of crossover trajectories is relatively high because the initial state is always the same. This is a strong assumption for the majority of the RL tasks/MDPs. Therefore, this has strongly limited the applicability of the proposed Graph Backup method.
>
> We agree that Graph Backup as currently formulated heavily relies on the exact crossover between trajectories to provide a benefit on top of Tree Backup or other methods. However, we think it’s possible to extend the method to settings where exact state matching may not occur, through the use of a similarity function, as is often done for count-based methods in exploration. While the current method doesn’t provide this, we still believe it’s a step in the right direction for more structure-aware backup methods for data-efficient reinforcement learning. We also note that other widely adopted methods have been proposed and accepted at major conferences solely on the back of results on the Atari benchmark [1,2,3,4,5].
>
> [1] Model-Based Reinforcement Learning for Atari, ICLR2019
>
> [2] Mastering Atari with Discrete World Models, ICLR2021
>
> [3] Data-Efficient Reinforcement Learning with Self-Predictive Representations, ICLR2021
>
> [4] Muesli: Combining Improvements in Policy Optimization, ICML2021
>
> [5] Multi-Game Decision Transformers, Neurips2022
>
> > It is not clear why a data graph is necessary for the proposed backup in Equation (6), and it looks like that this critical information is only included in the Appendix. Therefore it is very hard to evaluate the motivation of the proposed data graph.
>
> To clarify, in eq. 5  $\\gamma \\sum\_{a'} \\pi(a’|s\_t) G\_{\\hat{s}’}^{a'}$ we sum over all out-going actions starting from $\\hat{s}’$, which causes the state branching of Graph Backup as illustrated in Figure 1 (b). Eq. 5 is a recursive definition where one can further expand $G\_{\\hat{s}’}^{a'}$, which will increase the depth of the graph.
> On the other hand, we agree that the Algorithm listing for Double Distributional Graph Backup will also be helpful to understand how the algorithm works, since it explicitly iterates through transitions in Line 7. We’ve moved it to the main body of the paper.
>
> > In computing the new backup, there is a computational overhead since it requires more computations with the iterative update and multiple crossover trajectories. It would be good to add some discussions/evaluations of the computational overhead.
>
> Thanks for the suggestion. We’ve added a new section for computational overhead in the Appendix L. In brief, while there is an overhead for graph backup, this could be reduced substantially with a more efficient implementation. We also note that in general for data efficient RL we often need to make use of additional compute to make up for the limit on the number of environment transitions we’re allowed.
>
> > Section 4.2: This is more like a discussion. While it is good to have such kind of discussions, the paper would be stronger if it includes some formal results for the described advantages.
>
> Thanks for your suggestion. We add a new analysis about the value estimation of DQN on EmptyRoom with backup methods in Appendix M Figure 7. We also discussed why Graph Backup is helpful in the General response.
>
> > The writing of the paper needs significant work. While this is normally not the reason to reject a paper, it is making it really challenging to evaluate the paper itself when clarify is a serious issue.:
>
> Thanks for all the suggestions for improving clarity. We’ve implemented all the suggestions you made and also fixed the further grammatical errors with a grammar checker.
>
>
> Thanks for all your suggestions and comments. We hope we’ve addressed your criticisms, and if so, we’d appreciate you raising your score, given we think we’ve addressed weaknesses 1, 2 and 5 in your review.

---

> > ### Comment · Reviewer_zkNq · 2022-12-09
> > **Maintaining my score**
> >
> > Thanks the authors for their response. After reading the authors' responses and the other reviewers' comments, I'd like to keep my evaluation.

---

### Official Review · Reviewer_8RrV · 2022-10-24

**Confidence:** 3
**Correctness:** 4
**Technical Novelty And Significance:** 3
**Empirical Novelty And Significance:** 3
**Recommendation:** 6

**Clarity, Quality, Novelty And Reproducibility:**

The paper presents a novel graph backup algorithm and show consistent improvements on the experiments.

**Strength And Weaknesses:**

**Strength**

* The method is well-motivated and novel and can be combined with many existing off-policy algorithms.
* The empirical results are convincing and comprehensive (experiments on MiniGrid, MiniAtar, Atari100K).

**Weakness**:

* Some of the pseudocode is hard to follow (See my questions).
* The implementation of the graph is complicated.
* The applicability to continuous state space or more complex games with a high novel state ratio is unclear.
* Computation costs might be big to build and loop over the graph?

Questions:
* In Line 2, Algorithm 1, what does “the largest element” mean?
* Algorithm 2 in the Appendix is confusing. Where is $\bar{G}^{a}_s$ used in the update?
* Since graph backup performs recursive max-operation while looking ahead from a node, would graph backup aggravate over-estimation?
* I wonder when the novel state ratio is low (i.e., many crossover states), would a discrete graph planning achieve competitive performance with deep RL?


**Summary Of The Paper:**

This paper proposes a new target value estimate approach, Graph backup, based on tree-backup. The main idea is to leverage the crossover on state space to accelerate value propagation using a graph. In graph backup, a state-transition graph is built and the target value for each value backup is estimated by looking at the Q-values a few steps ahead on the graph. The empirical results show that graph backup can be a drop-in replacement to Rainbow (a DQN variant) and attains visible performance improvements, particularly in MiniGrid ( challenging sparse reward tasks).

**Summary Of The Review:**

Overall, I enjoy reading the paper and the analysis section in particular. The writing can be improved to make it easier to follow (especially the pseudocode). The algorithm has some novelty, but more work needs to be done in order to apply such a method to continuous state space.

---

> ### Author Response · Authors · 2022-11-17
> **Individual Response**
>
> Thanks for your support of our work and your constructive feedback! We try to address your concerns in the following, please let us know if there are further questions:
>
> > The implementation of the graph is complicated.
>
> We agree that the implementation of Graph Backup is more complicated than multi-step backup methods, but it can still be plugged into any algorithm that needs value estimation. For example, in `graphbackup/atari/src/gbsampler.py` we put all the logic relevant to Graph Backup into a single file, including graph building and backup.
>
> > Computation costs might be big to build and loop over the graph?
>
> The breath and depth limit can partially address the loop over the graph. On other hand, we argue that running Graph Backup is not more expensive than a model-based planning method that needs extra dynamics model prediction. We also add a new section discussing computational cost in Appendix L. In brief, while graph backup is computationally more expensive than one-step backup, this slowdown could be reduced by better-optimised code, and is in some sense necessary for improved performance: In the data efficient setting, given we’re limited to a certain number of samples, we have to leverage additional compute to improve performance.
>
> > Algorithm 2 in the Appendix is confusing. Where is $G^a_s$used in the update?
>
> $G^a_s$ is the formula for the non-distributional RL setting. The Graph Backup does one-step backups on the local transition graph. So the loop in Line 7 in the algorithm iterates over all the transitions in the transition graph and Line 8-Line 12 is doing a one-step C51 distributional backup.
>
> > In Line 2, Algorithm 1, what does “the largest element” mean?
>
> Sorry, that was a typo. It should be the last element, namely, `l[-1]`. We’ve corrected it.
>
> > Since graph backup performs recursive max-operation while looking ahead from a node, would graph backup aggravate over-estimation?
>
> We also worried about that and tried other variations, for example, the weighted average of future value estimations, but they are harmful to the performance. To further investigate the over-estimation problem, we add a new analysis of the value estimation in Appendix M Figure 7. We found that with online interaction, the value estimation of Graph Backup is actually more accurate than other backup methods and overestimation didn’t happen empirically. We also discussed why Graph Backup is helpful in the General response.
>
>
> > I wonder when the novel state ratio is low (i.e., many crossover states), would a discrete graph planning achieve competitive performance with deep RL?
>
> We tried Tabular Q learning for most of the tasks. It turns out Tabular Q learning can only work in very simple tasks like MiniGrid. The problem with tabular Q learning was it failed to decide how to act in the novel states and resorted to the uniform sampling of actions, leading to a random walk behaviour in MinAtar and Atari. To address that, we also tried to constantly distil the Q values of a tabular Q algorithm into a neural network, and use the Q network to initialise the Q values of the novel states. Such a hybrid approach is competitive to Graph Backup in Minigrid (mean 0.68) but still inferior in the MinAtar (mean 6.3).

---

> > ### Comment · Reviewer_8RrV · 2022-11-26
> > **Thanks for the response**
> >
> > Thanks the authors for their response. I am happy to keep my rating (6).

---

### Official Review · Reviewer_uNoE · 2022-10-25

**Confidence:** 4
**Correctness:** 2
**Technical Novelty And Significance:** 2
**Empirical Novelty And Significance:** 3
**Recommendation:** 5

**Clarity, Quality, Novelty And Reproducibility:**

Obtaining accurate bootstrapping targets is indeed essential in RL as most methods rely on this mechanism, either through value based methods or policy-based ones. Multistep TD methods generally only consider the sampled trajectory to perform updates, as a way to achieve scalability. My understanding is that the method the authors propose tries to construct the underlying graph of the MDP directly on observations, which inherently seems not scalable to large environments. By constructing this underlying graph, the method the authors propose to bootstrap from targets belonging to different possible trajectories, weighted by their probability of taking place. I am having a hard time understanding how this motivation differs from standard dynamic programming updates, yet there is no mention of it in the paper. Arguably, the authors do not try to bootstrap from all possible futures, only from the subset which has previously been observed. However, this strategy, based on modelling each observation as a node of the graph, faces the same curse of dimensionality. This is illustrated by the empirical results on Atari, which do not provide significant gain as the confidence intervals overlap, as well by how sparse the resulting graph is (.927 novel state ratio). I would really appreciated more discussion on this in the paper, in particular if the authors considered working in a (possibly discrete) latent space. For example, the authors mention that the transition matrix is estimate through counts. Are they also using counts for Atari, or instead are they using pseudo-counts? Moreover, how long does it take for the algorithm to perform updates, compared to other methods?

The authors claim that performing the Graph Backup update will lead to reduced variance. Although this is intuitive as expected updates usually do so (Expected SARSA or Expected Policy Gradients, which are not cited), I would have liked to see some experiments to support this.

Throughout the paper, there are some fundamental concepts that seem wrong. For example in section 3 "n-step-Q exploits the chain structure of the trajectories with little computational cost, but at a cost of biased target estimation". It is not clear what is meant by this. Q-Learning add bias by boostrapping, as do all TD methods when lambda is not set to 1.


Regarding experiments, the results on MinAtar and MiniGrid are good, however they are done with only 5 seeds which is very little considering how fast they can be performed. Moreover, the variance of the plots is very high, which brings into question the robustness of the method.

The paper adequately cites recent work such as Expected Eligibility Traces, however they do not provide a comparison to it in the experiments. There is in fact no comparison to any other method trying to performing credit assignment more efficiently, such as [1] or [2]. Recent work on leveraging the underlying structure of the graph for better information propagation would be a particularly good baseline [3]. In general I there isn't enough relevant baselines or other recent work on unifying multi step methods [4].


[1] Counterfactual Credit Assignment in Model-Free Reinforcement Learning.  Mesnard et al. 2020
[2] Hindsight Credit Assignment, Harutyunyan et al. 2019
[3] Reward propagation using graph convolutional networks, Klissarov et al. 2020
[4] Multi-step reinforcement learning: A unifying algorithm, De Asis et al., 2018

**Strength And Weaknesses:**

# Strengths
- The paper tackles an important problem
- Trying to construct graph to better perform credit assignment can be an promising path in RL
- Evaluation on a variety of environments

# Weaknesses
- The graph construction method seems inherently not scalable
- The results are not really statistically significant in larger environments
- Some claims should be more nuanced or supported by evidence

**Summary Of The Paper:**

The paper suggests that an important way to improve sample efficiency in reinforcement learning comes from bootstrapping from accurate target estimates. Although multistep methods such as TD(lambda) offer a bias-variance trade-off, the authors argue that such algorithms fail to exploit the underlying graph structure of the MDP. They therefore propose to explicitly build a graph structure out of the experience of the agent allowing for bootstrapping targets to be composed of various trajectories starting from the current state. The authors claim that this method leads to counterfactual credit assignment and reduced variance. They evaluate their method on variety of environments, from smaller grid-like worlds to the  Atari domain.

**Summary Of The Review:**

The proposed method tackles an important challenge in RL, but the claims are sometimes hardly justified. Importantly, the paper proposes an inherently non-scalable method, and it is not clear how it can avoid the curse of dimensionality. On some environments the propose method provides significant gain, but on larger ones it doesnt. More important, those results on only 5 seeds and the CIs are sometimes not even presented.

---

> ### Author Response · Authors · 2022-11-17
> **Individual Response (1)**
>
> Thanks for your constructive feedback. We noticed that your review is the same as one of the ICML reviews we got. As a kind reminder, some of the concerns you mentioned in the review might have been addressed. In particular, we believe the Atari results and section 6 can address some of your major concerns.
>
> > I am having a hard time understanding how this motivation differs from standard dynamic programming updates yet there is no mention of it in the paper.
>
> Tabular dynamic programming will not leverage neural network function estimation when doing a backup, while Graph Backup does. Also, dynamic programming (if you mean policy/value iteration) is designed for the case where transition dynamics are known. Graph Backup is more like the Search Tree Backup used by Alphazero on the transition graph, without a dynamics model.
>
>
> > This is illustrated by the empirical results on Atari, which do not provide significant gain as the confidence intervals overlap
>
> While there is a minor overlap in mean scores, that’s not the case for IQM, which is a metric with less variance.
>
> > I would really appreciated more discussion on this in the paper, in particular if the authors considered working in a (possibly discrete) latent space. For example, the authors mention that the transition matrix is estimate through counts. Are they also using counts for Atari, or instead are they using pseudo-counts?
>
> Thanks for the suggestion. Currently, we are using exact observation counts for Atari, which works surprisingly well. This work tries to keep things simple and show the power of Graph Backup without relying on an extra similarity kernel. While we agree that clustering or using a latent space/similarity kernel is a good direction and may further improve the performance, it will also introduce more components to the algorithm, which would make it more difficult to understand where performance benefits are coming from.
>
> >  as well by how sparse the resulting graph is (.927 novel state ratio)
>
> We discussed this in section 6 analysis: Atari Transition Graphs. The .927 novel state ratio doesn’t mean Graph Backup will always be the same as Tree Backup.
> For example, if we assume that duplicated states happen independently, this means that in 53% of the backup updates, there will be a crossover on the next 10 steps, which means the Graph Backup will give a different value estimate to multi-step methods more than half the time (using a depth of 10). We also show the exact topology of the generated Graphs in the same section.
>
> > Although this is intuitive as expected updates usually do so (Expected SARSA or Expected Policy Gradients, which are not cited), I would have liked to see some experiments to support this.
>
> We believe the variance reduction of Graph Backup is not the same as Expected SARSA and Expected Policy Gradients. Graph backup additionally reduces the variance of the learning updates by ensuring that the same state-action pair always has the same value estimate regardless of which trajectory in the replay buffer it was sampled from (see Section 4.2). For experiments, we show the variance of different backup methods on an offline dataset in Appendix C Figure 5. In Appendix M in the revision, we add a value estimation visualization from the whole training process, which demonstrates that Graph backup converges to correct value estimates more quickly than other methods.
>
> > Moreover, the variance of the plots is very high, which brings into question the robustness of the method.
>
> There is no evidence that Graph Backup is less robust than all other backup methods. For MiniGrid and MinAtar, the variance of Graph Backup is higher, but this is because the baselines perform consistently badly. For Atari, there is no obvious difference in terms of variance between different methods.
>
> > and the CIs are sometimes not even presented.
>
> In Appendix Table 3, the statistics of every single task are listed.

---

> > ### Author Response · Authors · 2022-11-17
> > **Individual Response (2)**
> >
> > > Throughout the paper, there are some fundamental concepts that seem wrong. For example in section 3 "n-step-Q exploits the chain structure of the trajectories with little computational cost, but at a cost of biased target estimation". It is not clear what is meant by this. Q-Learning add bias by boostrapping, as do all TD methods when lambda is not set to 1.
> >
> > We mean n-step-Q backup is biased because it will not converge to the correct Q value (even in the tabular case) but one-step backup, Tree Backup and Graph Backup will.  Please see [1] for details. In brief, in the off-policy case, the rewards after the first step are not necessarily from the current policy, and hence the update is off-policy without any correction, making the value estimate biased.
> >
> > [1] Understanding Multi-Step Deep Reinforcement Learning: A Systematic Study of the DQN Target, 2019
> >
> >
> > > The paper adequately cites recent work such as Expected Eligibility Traces, however they do not provide a comparison to it in the experiments. There is in fact no comparison to any other method trying to performing credit assignment more efficiently, such as [1] or [2]. Recent work on leveraging the underlying structure of the graph for better information propagation would be a particularly good baseline [3]. In general I there isn't enough relevant baselines or other recent work on unifying multi step methods [4].
> >
> > Thanks for your suggestion. While we agree these works are related conceptually and we’ve discussed them in the related work, these methods are not suitable for Data-Efficient and Return maximization learning. For example, Expected Eligibility Traces is for on-policy learning and is not open-sourced. Most of the methods in [4] are also for on-policy (or slightly off-policy) methods that try to optimize throughput rather than sample efficiency. One-step and N-step-Q learning is a quite standard backup for off-policy methods and especially for value-based methods, and hence makes sense as the direct point of comparison. Finally, Hindsight replay [2] is designed for goal-reaching settings, and so doesn’t obviously apply in the tasks we consider where we don’t have multiple goals. It seems likely that Hindsight replay could be combined with our method to have complementary performance gains.

---

> > > ### Comment · Reviewer_uNoE · 2022-11-18
> > > **Response to Author rebuttal**
> > >
> > > I would like to thank the authors for their rebuttal.
> > >
> > > Concerning the difference with dynamic programming, although no true transition dynamics is used explicitly in the proposed approach, it does leverage the notion of counts to estimate the transition probability. In this sense, the conceptual difference is not very large, and for this reason the method faces the same challenge of dimensionality. This is shown by the fact that there is only a 53% chance of crossover in 10 steps, which looks very low to me.
> > >
> > > I appreciate the experiments on variance reduction. The fact that it is reduced because the target value stays the same makes sense from the point of view of target networks as usually used in deep RL.
> > >
> > > Looking at Figure 3, the variance of the method is obviously higher on Minigrid and Minatar, so I am not sure I understand the comment of the authors. I would have appreciated an attempt at understand why this is the case. Moreover, the confidence intervals should be present in Table 1.
> > >
> > > For n-step Q learning, with the appropriate importance sampling ratios it is not clear to me why it would not converge. Have the authors shown that Graph Backup converges to an unbiased value? If so I have missed it.
> > >
> > > Regarding the re-use of an almost identical review, I would have written a completely different review if the paper had been significantly modified since last submission, but unfortunately I don't find this to be the case. For example a major concern for me was that only 5 seeds were used, yet there hasn't been any additional runs. Paired with the high variance in the results, this is a remaining concern.

---

> > > > ### Author Response · Authors · 2022-11-18
> > > > **Thanks and Further Response**
> > > >
> > > > Thanks for the quick response so that we have a chance for further clarification. About your concerns left:
> > > >
> > > > > Concerning the difference with dynamic programming, although no true transition dynamics is used explicitly in the proposed approach, it does leverage the notion of counts to estimate the transition probability.  In this sense, the conceptual difference is not very large, and for this reason, the method faces the same challenge of dimensionality.
> > > >
> > > > The key conceptual difference is dynamic programming on an Empirical MDP (you mentioned here) cannot do anything meaningful for state-action pairs that are not tried before but Graph Backup will query the $Q$ network to check if such state-action pair actually have a higher $Q$ value than the actual actions observed in the dataset. And if that's the case, the estimated $Q$ value, rather than the $Q$ values of observed actions will be propagated and used for backup.
> > > > On the other hand, we agree on the variation of dynamic programming you mentioned here is relevant to Graph Backup on a high level and I've mentioned it in (another) revision of the paper.
> > > >
> > > > > Looking at Figure 3, the variance of the method is obviously higher on Minigrid and Minatar, so I am not sure I understand the comment of the authors. I would have appreciated an attempt at understanding why this is the case.
> > > >
> > > > We think looking at the variance of the aggregated metrics can be misleading so we recommend checking Table 3 in the Appendix for the variance for each individual task. For MiniGrid, there are 4 tasks in that Graph Backup has a higher variance but it's not difficult to see all other methods basically achieve 0 performance on these tasks. And that's why the aggregated variance of Graph Backup may seem higher. And that's also the case for MinAtar, extra variance of Graph Backup comes from freeway-MinAtar where all other methods achieve a performance close to 0.
> > > >
> > > > > For n-step Q learning, with the appropriate importance sampling ratios it is not clear to me why it would not converge.
> > > >
> > > > Sure, we agree there might exist some importance sampling ratios to balance the off-policy reward sequences and value estimations.
> > > > But to return to the sentence reviewer criticizing: "n-step-Q exploits the chain structure of the trajectories with little computational cost, but at a cost of biased target estimation" We are talking about the vanilla n-step-Q backup and it's biased because the reward sequences are generated by behaviour policy that generates the data but the n-step-Q backup treats it as if they are reward sequences generated by the target policy to be evaluated.
> > > >
> > > > > For example a major concern for me was that only 5 seeds were used, yet there hasn't been any additional runs.
> > > >
> > > > I'm not sure whether after checking the individual results in Table 3, you still believe more seeds are needed for MiniGrid and Minatar experiments. If that's the case, we can add more runs for them. To us, it's quite clear that Graph Backup performs quite well on MiniGrid tasks while baselines don't walk at all for most of the tasks besides the simplest Empty Room, given limited interactions. As for MinAtar, Graph Backup performs way better on the freeway than baselines but for all four other tasks, it's similar to TreeBackup. Our main results are on Atari100K, which includes 26 tasks with much higher task diversity and the baseline Data Efficient Rainbow is developed and well-tuned in this setting. With this single component upgrade from n-step-Q to Graph Backup, IQM of human-normalised scores improves from 16 to 34 (mean from 28 to 50). We don't think any other components of Rainbow has such a large influence.
> > > >
> > > > > Regarding the re-use of an almost identical review, I would have written a completely different review if the paper had been significantly modified since last submission, but unfortunately I don't find this to be the case.
> > > >
> > > > We do understand why the review is used, basically, 1) it's a bit hard to identify what we've updated since most of them are about presentation and 2) we don't have a chance to rebuttal given ICML Phase 1 rejection, so we cannot clarify the questions and misinterpretation due to our presentation problems.
> > > > We hope our revision and clarification have addressed some of your concerns and if that's the case, it would be great if you can reconsider the score based on further information we provided.

---

### Author Response · Authors · 2022-11-17
**General Response**

We thank all the reviewers for the thoughtful and constructive feedback.
To summarise positive feedback:
* All the reviewers think the experiments are comprehensive.
* Reviewer 8RrV, zkNq and iGcV think the empirical results are strong and promising
* Reviewer 8RrV and iGcV like the fact that Graph Backup is a single component that can be plugged into any existing off-policy RL methods.
* Reviewer 8RrV, zkNq thinks the method is well motivated

Here we address common concerns from reviewers:

### Intuitions about why Graph Backup is better than Tree Backup

We revised section 4.2 hoping to make the intuitions behind Graph Backup clearer. But here we also present a high-level intuition: the power of bootstrapped value estimation comes from the value estimation of future states and rewards in observed transitions. Multi-step backup is more efficient than one-step backup because it brings the value estimations of multiple future states and rewards in a segment of the trajectories into consideration. Graph Backup instead aggregates information from a subgraph of the whole transition graph, involving value estimations of even more possible future states and rewards in possible paths. The intuition behind this is actually similar to AlphaZero, where a tree of future states is expanded by a model and a powerful value estimator is built based on the tree. While Graph Backup is different in the sense that it doesn’t need a learned accurate model or a simulator.

To provide more intuition about Graph Backup, we add a new analysis in the **revised Appendix M**. We visualized how the value estimations of all the states for an empty room environment are updated along with the training, showing how Graph Backup propagate values quickly and stably to all the states without overestimation.

### Why Graph Backup works so well on Atari?

We agree with several of the reviewers that the increased performance on Atari is surprising, given we might intuitively think that there will be very few crossovers for these games. Here we give some intuition as to why Graph Backup still works well in this setting.

First, quoting from the paper:

> The average novel state ratio of Atari is 0.927, which means the graphs are usually quite sparse (The average novel state ratios of MiniGrid/MinAtar are 0.006/0.298 respectively). However, if we assume that duplicated states happen independently, this means that in 53% of the backup updates, there will be a crossover on the next 10 steps, which means the Graph Backup will give a different value estimate to multi-step methods more than half the time.

Second, consider that, from the transition graph visualisation in Figure 4 and the appendix, most transition graphs look roughly like (collections of) trees, with branches frequently sprouting but rarely reconnecting. This is due to the fixed starting state for Atari games, which will mean that many trajectories in the replay buffer will share a common prefix of transitions with many other trajectories in the replay buffer. For state-action pairs in these common prefixes, the variance of the value estimate will be much lower for Graph Backup than for other methods, where the value estimate will depend on which trajectory the transition is sampled from. These states are likely also a large portion of the replay buffer, and hence improving the value estimation on these states will lead to improved performance quicker than other methods. Note that this improvement doesn’t rely on trajectories crossing over again after they’ve split; in fact, even if the entire graph was a tree, we’d still expect to see the reduced variance in the value target due to this property for Graph Backup relative to other methods.

---

### Decision · Program_Chairs · 2023-01-20

**Decision:**

Reject

**Justification For Why Not Higher Score:**

Although the empirical results were promising, the reviewers did not understand how they came out of the paper's proposed approach.  (Additionally, one reviewer did not consider the empirical improvements significant).  None of the reviewers (including the one marginally positive reviewer) were willing to advocate for accepting the paper.


**Justification For Why Not Lower Score:**

n/a

**Metareview: Summary, Strengths And Weaknesses:**

(a) Summary: This work proposes to improve sample efficiency in RL by constructing bootstrapping targets from multiple trajectories starting at the current state, in a manner akin to tree backup.

(b) Strengths:  All the reviewers agreed that the paper studies an important problem.  Exploiting graph structure in the environment's dynamics may be a promising direction.  The empirical results seem promising.

(c) Weaknesses: All reviewers had significant clarity concerns.  It was not clear to multiple reviewers where the empirical improvements came from.  All reviewers agreed that this approach is likely to have significant scalability issues.


**Summary Of Ac-Reviewer Meeting:**

n/a